# Uncertainty Estimation and Quantification for LLMs: A Simple Supervised Approach

## Abstract

In this paper, we study the problem of uncertainty estimation and calibration for LLMs. We begin by formulating the uncertainty estimation problem, a relevant yet underexplored area in existing literature. We then propose a supervised approach that leverages labeled datasets to estimate the uncertainty in LLMs' responses. Based on the formulation, we illustrate the difference between the uncertainty estimation for LLMs and that for standard ML models and explain why the hidden neurons of the LLMs may contain uncertainty information. Our designed approach demonstrates the benefits of utilizing hidden activations to enhance uncertainty estimation across various tasks and shows robust transferability in out-of-distribution settings. We distinguish the uncertainty estimation task from the uncertainty calibration task and show that better uncertainty estimation leads to better calibration performance. Furthermore, our method is easy to implement and adaptable to different levels of model accessibility including black box, grey box, and white box.

## 1 Introduction

Large language models (LLMs) have marked a significant milestone in the advancement of natural language processing (Radford et al., 2019; Brown et al., 2020; Ouyang et al., 2022; Bubeck et al., 2023), showcasing remarkable capabilities in understanding and generating human-like text. However, their tendency to produce hallucinations—misleading or fabricated information—raises concerns about their reliability and trustworthiness (Rawte et al., 2023). The problem of whether we should trust the response from machine learning models is critical in machine-assisted decision applications, such as self-driving cars (Ramos et al., 2017), medical diagnosis (Esteva et al., 2017), and loan approval processes (Burrell, 2016), where errors can lead to significant loss.

This issue becomes even more pressing in the era of generative AI, as the outputs of these models are random variables sampled from a distribution, meaning incorrect responses can still be produced with positive probability. Due to this inherent randomness, the need to address uncertainty estimation in generative AI is even greater than that in other machine learning models (Gal & Ghahramani, 2016; Lakshminarayanan et al., 2017; Guo et al., 2017; Minderer et al., 2021), and yet there has been limited research in this area (Kuhn et al., 2023; Manakul et al., 2023; Tian et al., 2023).

In this work, we aim to formally define the problem of uncertainty estimation for LLMs and propose methods to address it. As shown in Figure 1, uncertainty estimation for LLMs can be broadly defined as the task of predicting the quality of the generated response based on the input. In this context, "quality" typically refers to aspects such as confidence, truthfulness, and uncertainty. Assuming access to a universal metric for evaluating the confidence of the output, the goal of uncertainty estimation is to produce a confidence score that closely aligns with this metric. Given the inherent randomness in LLMs, where incorrect responses can still be generated with positive probability, uncertainty estimation serves as a crucial safeguard. It helps assess the reliability of responses, enhance the trustworthiness of the model, and guide users on when to trust or question the output.

It is also worth noting that calibration is closely related and can be viewed as a subclass of uncertainty estimation, where the metric corresponds to the conditional probability in the individual level. Most studies on uncertainty estimation or calibration in language models focus on fixed-dimensional prediction tasks (i.e., the output of the LLM only has one token limited in a finite set), such as sentiment analysis, natural language inference, and commonsense reasoning (Zhou et al., 2023; Si et al.,

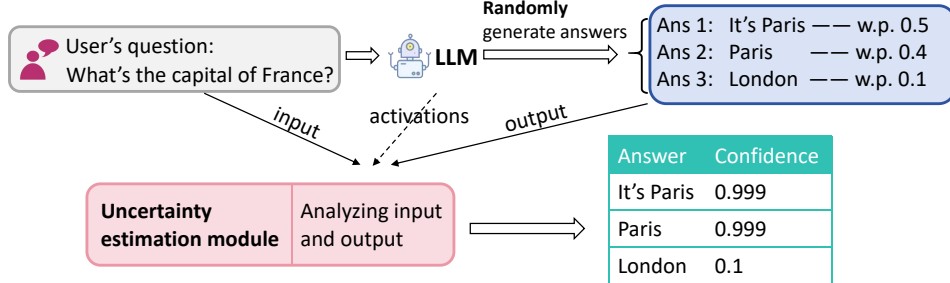

Figure 1: An example to illustrate the uncertainty estimation task. The LLM randomly generates an answer to the question (It's Paris, Paris, or London). The goal of the uncertainty estimation is to estimate a confidence score to the question-answer pair, where a higher score indicates a higher confidence to believe that the answer is correct.

2022; Xiao et al., 2022; Desai & Durrett, 2020). However, given the structural differences in how modern LLMs are used, alongside their proven capability to handle complex, free-form tasks with variable-length outputs, there is a growing need to address uncertainty estimation and calibration specifically for general language tasks in the domain of LLMs.

This work explores a simple supervised method motivated by two ideas in the existing literature on LLMs. First, prior work on uncertainty estimation for LLMs primarily focused on designing uncertainty metrics in an unsupervised way by examining aspects like the generated outputs' consistency, similarity, entropy, and other relevant characteristics (Lin et al., 2023; Manakul et al., 2023; Kuhn et al., 2023; Hou et al., 2023; Lin et al., 2022; Kuhn et al., 2023; Chen et al., 2024). The absence of the need for knowledge of the model's weights enables their application to some black-box or gray-box models. Second, a growing stream of literature argues that hidden layers' activation values within the LLMs offer insights into the LLMs' knowledge and confidence (Slobodkin et al., 2023; Ahdritz et al., 2024; Duan et al., 2024). It has shown success in other fields of LLMs, like hallucination detection (CH-Wang et al., 2023; Azaria & Mitchell, 2023; Ahdritz et al., 2024). Based on this argument, white-box LLMs, which allow access to more of LLMs' inner values, such as logits and hidden layers, are believed to have the capacity to offer a more nuanced understanding and improved uncertainty estimation results (Verma et al., 2023; Chen et al., 2024; Plaut et al., 2024).

Both of the above approaches, however, have key limitations. For the unsupervised metrics, given the complexity of LLMs' underlying architectures, semantic information may be diluted when processing through self-attention mechanisms and during token encoding/decoding. For the second idea, the requirements of hidden layer features restrict its application to close-source/black-box LLMs. In this paper, we combine the strengths of these two ideas by proposing a general supervised learning method and pipeline design that address these limitations. Specifically, to incorporate more features (e.g., hidden layers) in estimating the uncertainty, we train an external uncertainty estimation model in a supervised way to estimate the uncertainty/confidence of the response generated from an LLM (*target LLM*). As the quality of the response reveals to what extent we should believe the response is correct, we formulate this supervised uncertainty estimation problem as a regression task and prepare the labels in the training dataset by measuring the response's quality. To extend our method to black-box LLMs, we allow the semantic features of the question-response pair to come from another language model (*tool LLM*). The overall pipeline of this method is shown in Figure 2.

Our contributions are four-fold:

- First, we formally define the task of uncertainty estimation, while some of the existing literature either does not distinguish uncertainty estimation and uncertainty calibration or misuses and confuses the terminologies of uncertainty and hallucination.

- Second, we adopt a supervised method for uncertainty estimation that is intuitive, easy to implement, and executable even on black-box LLMs. Leveraging supervised labels from the uncertainty metric, our approach sets an upper bound for the performance of all unsupervised methods, representing the highest achievable performance for these approaches.

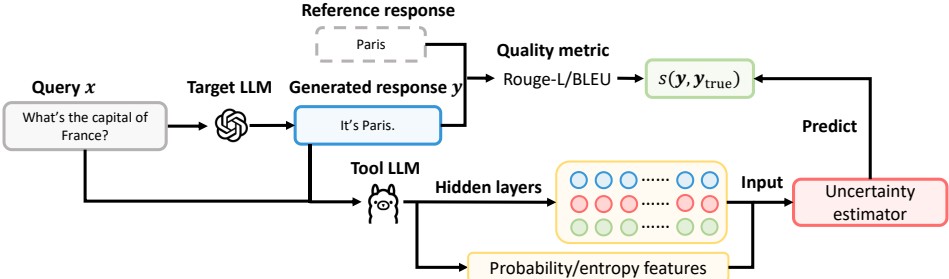

Figure 2: Illustration of our proposed supervised method. The tool LLM is an open-source LLM and can be different from the target LLM. In the training phase, where the reference response is available, we train the uncertainty estimator using the quality of the response as the label. In the test phase, the uncertainty estimator predicts the quality of the generated response to obtain an uncertainty score.

- Third, we systematically discuss the relationship and the difference between deep learning and LLM in uncertainty estimation. Formally, we give an explanation to see why the method for the traditional deep learning model may fail in LLM, and why the hidden layer is useful in estimating the uncertainty in our context.

- Finally, numerical experiments on various natural language processing tasks demonstrate the superiority of our methods over existing benchmarks. The results also reveal several insightful observations, including the role of neural nodes in representing uncertainty, and the transferability of our trained uncertainty estimation model.

## 1.1 RELATED LITERATURE

The uncertainty estimation and calibration for traditional machine learning is relatively well-studied (Abdar et al., 2021; Gawlikowski et al., 2023). However, with the rapid development of LLMs, there is a pressing need to better understand the uncertainty for LLMs' responses, and measuring the uncertainty from sentences instead of a fixed-dimension output is more challenging. One stream of work has been focusing on unsupervised methods that leverage entropy (Malinin & Gales, 2021), similarity (Fomicheva et al., 2020; Lin et al., 2022), semantic (Kuhn et al., 2023; Duan et al., 2023), logit or hidden states' information (Kadavath et al., 2022; Chen et al., 2024; Su et al., 2024; Plaut et al., 2024) to craft an uncertainty metric that helps to quantify uncertainty. For black-box models, some of the metrics can be computed based on multiple sampled output of the LLMs (Malinin & Gales, 2021; Lin et al., 2023; Manakul et al., 2023; Chen & Mueller, 2023); while for white-box models, more information such as the output's distribution, the value of the logit and hidden layers make computing the uncertainty metric easier. We also refer to Desai & Durrett (2020); Zhang et al. (2021); Ye & Durrett (2021); Si et al. (2022); Quach et al. (2023); Kumar et al. (2023); Mohri & Hashimoto (2024) for other related uncertainty estimation methods such as conformal prediction. We defer more discussions on related literature, in particular, on the topics of hallucination detection and information in hidden layers of LLMs, to Appendix A.

## 2 PROBLEM SETUP

Consider the following environment where one interacts with LLMs through prompts and responses: An LLM is given with an input prompt $\boldsymbol{x} = (x_1, x_2, ..., x_k) \in \mathcal{X}$ with $x_i \in \mathcal{V}$ representing the $i$-th token of the prompt. Here $\mathcal{V}$ denotes the vocabulary for all the tokens. Then the LLM randomly generates its response $\boldsymbol{y} = (y_1, y_2, ..., y_m) \in \mathcal{Y}$ following the probability distribution

$$y_j \sim p_\theta(\cdot|\boldsymbol{x}, y_1, y_2, ..., y_{j-1}).$$

Here the probability distribution $p_\theta$ denotes the distribution (over vocabulary $\mathcal{V}$) as the LLM's output, and $\theta$ encapsulates all the parameters of the LLM. The conditional part includes the prompt $\boldsymbol{x}$ and all the tokens $y_1, y_2, ..., y_{j-1}$ generated preceding the current position.

We consider using the LLM for some downstream NLP tasks such as question answering, multiple choice, and machine translation. Such a task usually comes with an evaluation/scoring function that

evaluates the quality of the generated response $s(\cdot, \cdot) : \mathcal{Y} \times \mathcal{Y} \to [0, 1]$. For each pair of $(\boldsymbol{x}, \boldsymbol{y})$, the evaluation function rates the response $\boldsymbol{y}$ with the score $z := s(\boldsymbol{y}, \boldsymbol{y}_{\text{true}})$ where $\boldsymbol{y}_{\text{true}}$ is the true response for the prompt $\boldsymbol{x}$. The true response $\boldsymbol{y}_{\text{true}}$ is usually decided by factual truth, humans, or domain experts, and we can assume it follows a distribution condition on the prompt $\boldsymbol{x}$. It does not hurt to assume a larger score represents a better answer; $z = 1$ indicates a perfect answer, while $z = 0$ says the response $\boldsymbol{y}$ is off the target.

We define the task of *uncertainty estimation* for LLMs as the learning of a function $g$ that predicts the score

$$g(\boldsymbol{x}, \boldsymbol{y}) \approx \mathbb{E}\left[s(\boldsymbol{y}, \boldsymbol{y}_{\text{true}})|\boldsymbol{x}, \boldsymbol{y}\right] \tag{1}$$

where the expectation on the right-hand side is taken with respect to the (possible) randomness of the true response $\boldsymbol{y}_{\text{true}}$, and for notational clarity, we omit the dependence of $\boldsymbol{y}_{\text{true}}$ on $\boldsymbol{x}$. We emphasize two points on this task definition: The uncertainty function $g$ takes the prompt $\boldsymbol{x}$ and $\boldsymbol{y}$ as its inputs. This implies (i) the true and predicted uncertainty score can and should depend on the specific realization of the response $\boldsymbol{y}$, not just $\boldsymbol{x}$ (Zhang et al., 2021; Kuhn et al., 2023), and (ii) the uncertainty function $g$ does not require the true response $\boldsymbol{y}_{\text{true}}$ as the input.

We note that a significant body of literature explores uncertainty estimation and calibration in language models (Zhou et al., 2023; Si et al., 2022; Xiao et al., 2022; Desai & Durrett, 2020). They primarily focus on classification tasks where outputs are limited to a finite set of tokens (i.e., $\boldsymbol{y}$ contains only one element). In contrast, our work extends this to allow free-form responses, and the ability to handle variable-length outputs aligns more closely with current advancements in LLMs.

# 3 UNCERTAINTY ESTIMATION VIA SUPERVISED LEARNING

## 3.1 OVERVIEW OF SUPERVISED UNCERTAINTY ESTIMATION

We consider a supervised approach of learning the uncertainty function $g : \mathcal{X} \times \mathcal{Y} \to [0, 1]$, which is similar to the standard setting of uncertainty quantification for ML/deep learning models. First, we start with a raw dataset of $n$ samples

$$\mathcal{D}_{\text{raw}} = \{(\boldsymbol{x}_i, \boldsymbol{y}_i, \boldsymbol{y}_{i,\text{true}}, s(\boldsymbol{y}_i, \boldsymbol{y}_{i,\text{true}}))\}_{i=1}^n .$$

$\mathcal{D}_{\text{raw}}$ can be generated based on a labeled dataset for the tasks we consider. Here $\boldsymbol{x}_i = (x_{i,1}, ..., x_{i,k_i})$ and $\boldsymbol{y}_i = (y_{i,1}, ..., y_{i,m_i})$ denote the prompt and the corresponding LLM's response, respectively. $\boldsymbol{y}_{i,\text{true}}$ denotes the true response (that comes from the labeled dataset) of $\boldsymbol{x}_i$, and $s(\boldsymbol{y}_i, \boldsymbol{y}_{i,\text{true}})$ assigns a score for the response $\boldsymbol{y}_i$ based on the true answer $\boldsymbol{y}_{i,\text{true}}$.

The next is to formulate a supervised learning task based on $\mathcal{D}_{\text{raw}}$. Specifically, we construct

$$\mathcal{D}_{\text{sl}} = \{(\boldsymbol{v}_i, z_i)\}_{i=1}^n$$

where $z_i := s(\boldsymbol{y}_i, \boldsymbol{y}_{i,\text{true}}) \in [0, 1]$ denotes the target score to be predicted. The vector $\boldsymbol{v}_i$ summarizes useful features for the $i$-th sample based on $(\boldsymbol{x}_i, \boldsymbol{y}_i)$. With this design, a supervised learning task on the dataset $\mathcal{D}_{\text{sl}}$ coincides exactly with learning the uncertainty estimation task defined in (1).

**Getting Features.** When constructing $\boldsymbol{v}_i$, a natural implementation is to use the features of $(\boldsymbol{x}, \boldsymbol{y})$ extracted from the LLM (denoted as *target LLM*) that generates the response $\boldsymbol{y}$ as done in Duan et al. (2024) for hallucination detection and Burns et al. (2022) for discovering latent knowledge. This method functions effectively with white-box LLMs where hidden activations are accessible. We note that obtaining hidden layers' activations merely requires an LLM and the prompt-response pair $(\boldsymbol{x}, \boldsymbol{y})$, and the extra knowledge of uncertainty can come from the hidden layers of any white-box LLM that takes as input the $(\boldsymbol{x}, \boldsymbol{y})$ pair, not necessarily from the target LLM.

Another note is that our goal is to measure the uncertainty of the input-output pair $(\boldsymbol{x}, \boldsymbol{y})$ using the given metric, which is independent of the target LLM that generates the output from input $\boldsymbol{x}$. Therefore, due to the unique structure of LLMs, any white-box LLM can take $(\boldsymbol{x}, \boldsymbol{y})$ together as input, allowing us to extract features from this white-box LLM (referred to as the *tool LLM*).

This observation has two implications: First, if the *target LLM* is a black-box one, we can rely on a white-box *tool LLM* to extract feature; Second, even if the *target LLM* is a Which-box one, we can also adopt a more powerful white-box *tool LLM*) that could potentially generate more useful feature. In Algorithm 1, we present the algorithm of our pipeline that is applicable to *target LLMs* of any type, and we provide an illustration of the algorithm pipeline in Figure 2.

---

**Algorithm 1** Supervised uncertainty estimation

---

**Input:** Target LLM $p_\theta$ (the uncertainty of which is to be estimated), tool LLM $q_\theta$ (used for uncertainty estimation), a labeled training dataset $\mathcal{D}$, a test sample with prompt $x$
1: %% Training phase:
2: Use $p_\theta$ to generate responses for the samples in $\mathcal{D}$ and construct the dataset $\mathcal{D}_{\text{raw}}$
3: For each sample $(x_i, y_i) \in \mathcal{D}_{\text{raw}}$, extract features (hidden-layer activations, entropy- and probability-related features) using the LLM $q_\theta$, and then construct the dataset $\mathcal{D}_{\text{sl}}$
4: Train a supervised learning model $\hat{g}$ that predicts $z_i$ with $v_i$ based on the dataset $\mathcal{D}_{\text{sl}}$
5: %% Test phase:
6: Generate the response $y$ for the test prompt $x$
7: Extract features $v$ using $q_\theta$
**Output:** Associate the response $y$ with the uncertainty score $\hat{g}(v)$

---

### 3.2 FEATURES FOR UNCERTAINTY ESTIMATION

A bunch of features that can be extracted from an LLM show a potential relationship to the measurement of uncertainty in the literature. Here we categorize these features into two types based on their sources:

*White-box features:* LLM's hidden-layer activations. We feed $(x_i, y_i)$ as input into the tool LLM, and extract the corresponding hidden layers' activations of the LLM.

*Grey-box features:* Entropy- or probability-related outputs. The entropy of a discrete distribution $p$ over the vocabulary $\mathcal{V}$ is defined by $H(p) := -\sum_{v \in \mathcal{V}} p(v) \log(p(v))$. For a prompt-response pair $(x, y) = (x_1, ..., x_k, y_1, ..., y_m)$, we consider as the features the entropy at each token such as $H(q_\theta(\cdot | x_1, ..., x_{j-1}))$ and $H(q_\theta(\cdot | x, y_1, ..., y_{j-1}))$ where $q_\theta$ denotes the tool LLM. We defer the detailed discussions on feature construction to Appendix D.

There can be other useful features such as asking the LLM "how certain it is about the response" (Tian et al., 2023). We do not try to exhaust all the possibilities, and the aim of our paper is more about formulating the uncertainty estimation for the LLMs as a supervised task and understanding how the internal states of the LLM encode uncertainty. To the best of our knowledge, our paper is the first one to do so. Specifically, the above formulation aims for the following two outcomes: (i) an uncertainty model $\hat{g}(v_i)$ that predicts $z_i$ and (ii) knowing whether the hidden layers carry the uncertainty information.

### 3.3 THREE REGIMES OF SUPERVISED UNCERTAINTY ESTIMATION

In Section 3.1, we present that our supervised uncertainty estimation method can be extended to a black-box LLM by separating the target LLM and tool LLM. Next, we formally present our method for white-box, grey-box, and black-box target LLMs.

**White-box supervised** uncertainty estimation (Wb-S): This Wb-S approach is applicable to a white-box LLM where the tool LLM coincides with the target LLM (i.e., $p_\theta = q_\theta$).

**Grey-box supervised** uncertainty estimation (Gb-S): This Gb-S regime also uses the same target and tool LLMs ($p_\theta = q_\theta$) and constructs the features only from the grey-box source, that is, those features relying on the probability and the entropy (such as those in Table 5 in Appendix D), but it ignores the hidden-layer activations.

**Black-box supervised** uncertainty estimation (Bb-S): The Bb-S regime does not assume the knowledge of the parameters of $p_\theta$ but still aims to estimate its uncertainty. To achieve this, it considers another open-source LLM denoted by $q_\theta$. The original data $\mathcal{D}_{\text{raw}}$ is generated by $p_\theta$ but then the uncertainty estimation data $\mathcal{D}_{\text{sl}}$ is constructed based on $q_\theta$ from $\mathcal{D}_{\text{raw}}$ as illustrated in the following diagram

$$\mathcal{D}_{\text{raw}} \xrightarrow{q_\theta} \mathcal{D}_{\text{sl}}.$$

For example, for a prompt $x$, a black-box LLM $p_\theta$ generates the response $y$. We utilize the open-source LLM $q_\theta$ to treat $(x, y)$ jointly as a sequence of (prompt) tokens and extract the features of hidden activations and entropy as in Section 3.2. In this way, we use $q_\theta$ together with the learned

uncertainty model from $\mathcal{D}_{\mathrm{sl}}$ to estimate the uncertainty of responses generated from $p_\theta$ which we do not have any knowledge about.

## 4 INSIGHTS FOR THE ALGORITHM DESIGN

### 4.1 UNCERTAINTY ESTIMATION V.S. UNCERTAINTY CALIBRATION

So far in this paper, we focus on the uncertainty estimation task which aims to predict the quality of the response to reveal whether the LLM makes mistakes in its response or not. There is a different but related task known as the uncertainty calibration problem. In comparison, the uncertainty calibration aims to ensure that the output from the uncertainty estimation model for (1) conveys a probabilistic meaning. That is, $g(\boldsymbol{x}, \boldsymbol{y})$ is defined as the *probability* that $\boldsymbol{y}$ is true. This is compatible with our method by replacing the quality $s(\boldsymbol{y}, \boldsymbol{y}_{\mathrm{true}})$ with $1\{\boldsymbol{y} \in \mathcal{Y}_{\mathrm{true}}\}$, where $\mathcal{Y}_{\mathrm{true}}$ is a set containing all the possible true responses. Another aspect of the relation between our uncertainty estimation method and uncertainty calibration is that our method can be followed by any recalibration methods for ML models to form a pipeline for calibration. And intuitively, a better uncertainty estimation/prediction will lead to a better-calibrated uncertainty model, which is also verified in our numerical experiments in Appendix C.

### 4.2 WHY HIDDEN LAYERS AS FEATURES?

In this subsection, we provide a simple theoretical explanation for why the hidden activations of the LLM can be useful in uncertainty estimation. Consider a binary classification task where the features $\boldsymbol{X} \in \mathbb{R}^d$ and the label $Y \in \{0, 1\}$ are drawn from a distribution $\mathcal{P}$. We aim to learn a model $f : \mathbb{R}^d \to [0, 1]$ that predicts the label $Y$ from the feature vector $\boldsymbol{X}$, and the learning of the model employs a loss function $l(\cdot, \cdot) : [0, 1] \times [0, 1] \to \mathbb{R}$.

**Proposition 4.1.** *Let $\mathcal{F}$ be the class of measurable function that maps from $\mathbb{R}^d$ to $[0, 1]$. Under the cross-entropy loss $l(y, \hat{y}) = y \log(\hat{y}) + (1 - y) \log(1 - \hat{y})$, the function $f^*$ that minimizes the loss*

$$f^* = \arg\min_{f \in \mathcal{F}} \mathbb{E}\left[l(Y, f(\boldsymbol{X}))\right]$$

*is the Bayes optimal classifier $f^*(\boldsymbol{x}) = \mathbb{P}(Y = 1|\boldsymbol{X} = \boldsymbol{x})$ where the expectation and the probability are taken with respect to $(\boldsymbol{X}, Y) \sim \mathcal{P}$. Moreover, the following conditional independence holds*

$$Y \perp \boldsymbol{X} \mid f^*(\boldsymbol{X}).$$

The proposition is not technical and it can be easily proved by using the structure of $f^*(\boldsymbol{X})$ so we refer the proof to Berger (2013). It states a nice property of the cross-entropy loss that the function learned under the cross-entropy loss coincides with the Bayes optimal classifier. Note that this is contingent on two requirements. First, the function class $\mathcal{F}$ is the measurable function class. Second, it requires the function $f^*$ learned through the population loss rather than the empirical loss/risk. The proposition also states one step further on conditional independence $Y \perp \boldsymbol{X} \mid f^*(\boldsymbol{X})$. This means all the information related to the label $Y$ that is contained in $\boldsymbol{X}$ is summarized in the prediction function $f^*$. This intuition suggests that for classic uncertainty estimation problems, when a prediction model $\hat{f} : \mathbb{R}^d \to [0, 1]$ is well-trained, the predicted score $\hat{f}(\boldsymbol{X})$ should capture all the information about the true label $Y$ contained in the features $\boldsymbol{X}$, without relying on the features of $\boldsymbol{X}$. This indeed explains why the classic uncertainty estimation and calibration methods only work with the predicted score $\hat{f}(\boldsymbol{X})$ for re-calibration, including Platt scaling (Platt et al., 1999), isotonic regression (Zadrozny & Elkan, 2002), temperature scaling (Guo et al., 2017), etc.

When it comes to uncertainty estimation for LLMs, which is different from calibration and LLMs' structure is much more complex, we will no longer have conditional independence, and that requires additional procedures to retrieve more information on $Y$. The following supporting corollary states that when the underlying loss function $\tilde{l}$ does not possess this nice property (the Bayes classifier minimizes the loss point-wise) of the cross-entropy loss, the conditional independence will collapse.

**Corollary 4.2.** *Suppose the loss function $\tilde{l}$ satisfies*

$$\mathbb{P}\left(f^*(\boldsymbol{x}) \neq \arg\min_{\tilde{y} \in [0,1]} \mathbb{E}\left[\tilde{l}(Y, \tilde{y})|\boldsymbol{X} = \boldsymbol{x}\right]\right) > 0,$$

*where $f^*$ is defined as Proposition 4.1, then for the function $\tilde{f} = \arg\min_{f \in \mathcal{F}} \mathbb{E}\left[\tilde{l}(Y, f(\boldsymbol{X}))\right]$, where the expectation is with respect to $(\boldsymbol{X}, Y) \sim \mathcal{P}$, there exists a distribution $\mathcal{P}$ such that the conditional independence no longer holds*

$$Y \not\perp \boldsymbol{X} \mid \tilde{f}(\boldsymbol{X}).$$

Proposition 4.1 and Corollary 4.2 together illustrate the difference between uncertainty estimation for a traditional ML model and that for LLMs. In this task, the output $\tilde{f}(\boldsymbol{X})$ of the model (traditional ML model or LLM) is restricted in [0,1] to indicate the confidence of $Y = 1$. For the traditional ML models, the cross-entropy loss, which is commonly used for training the model, is aligned toward the uncertainty calibration objective. When it comes to uncertainty estimation for LLMs, the objective can be different from calibration, and the LLMs are often pretrained with some other loss functions (for example, the negative log-likelihood loss for next-token prediction) on diverse language tasks besides binary classifications. These factors cause a misalignment between the model pre-training and the uncertainty estimation task. Consequently, the original features (e.g., the output logits) may and should (in theory) contain information about the uncertainty score $Y$ that cannot be fully captured by $\tilde{f}(\boldsymbol{X})$. This justifies why we formulate the uncertainty estimation task as the previous subsection and take the hidden-layer activations as features to predict the uncertainty score; it also explains why we do not see much similar treatment in the mainstream uncertainty estimation literature (Kuhn et al., 2023; Manakul et al., 2023; Tian et al., 2023).

## 5 NUMERICAL EXPERIMENTS AND FINDINGS

### 5.1 LLMS, TASKS, BENCHMARKS, AND PERFORMANCE METRICS

Here we outline the general setup of the numerical experiments. Certain tasks may deviate from the general setup, and we will detail the specific adjustments as needed.

**LLMs.** For our numerical experiments, we mainly consider three open-source LLMs, LLaMA2-7B (L-7B) (Touvron et al., 2023), LLaMA3-8B (L-8B)(AI@Meta, 2024) and Gemma-7B (G-7B) (Gemma Team et al., 2024) as $p_\theta$ defined in Section 2. For certain experiments, we also employ the models of LLaMA2-13B and Gemma-2B. We also use their respective tokenizers as provided by Hugging Face. We do not change the parameters/weights $\theta$ of these LLMs.

**Tasks and Datasets.** We mainly consider three tasks for uncertainty estimation, question answering (the CoQA and TriviaQA (Joshi et al., 2017) datasets), multiple choice (the MMLU dataset (Hendrycks et al., 2020)), and machine translation (the WMT 2014 dataset (Bojar et al., 2014)). All the labeled datasets for these tasks are in the form of $\{(\boldsymbol{x}_i, \boldsymbol{y}_{i,\text{true}})\}_{i=1}^n$ where $\boldsymbol{x}_i$ can be viewed as the prompt for the $i$-th sample and $\boldsymbol{y}_{i,\text{true}}$ the true response. We adopt the few-shot prompting when generating the LLM's response $\boldsymbol{y}_i$, and we use 5 examples in the prompt of the multiple-choice task and 3 examples for the remaining natural language generation tasks. This enables the LLM's in-context learning ability (Radford et al., 2019; Zhang et al., 2023) and ensures the LLM's responses are in a desirable format. We defer more details of the few-shot prompting to Appendix D.1.

**Benchmarks**. We compare our approach with a number of the state-of-the-art benchmarks for the problem. Manakul et al. (2023) give a comprehensive survey of the existing methods and compare four distinct measures for predicting sentence generation uncertainty. The measures are based on either the maximum or average values of entropy or probability across the sentence, including Max Likelihood, Avg Likelihood, Max Ent, and Avg Ent (denoted as MaxL, AvgL, MaxE, AvgE) defined in Table 5. We note that each of these measures can be applied as a single uncertainty estimator, and they are all applied in an unsupervised manner that does not require additional supervised training. In particular, in applying these measures for the MMLU dataset, since the answer only contains one token from {A, B, C, D}, we use the probabilities and the entropy (over these four tokens) as the benchmarks which represent the probability of the most likely choice and the entropy of all choices, respectively. Kuhn et al. (2023) generate multiple answers, compute their entropy in a semantic sense, and define the quantity as *semantic entropy*. This semantic-entropy uncertainty (SU) thus can be used as an uncertainty estimator for the LLM's responses. Tian et al. (2023) propose the approach of asking the LLM for its confidence (denoted as A4U) which directly obtains the uncertainty score from the LLM itself.

**Our methods.** We follow the discussions in Section 3.3 and implement three versions of our proposed supervised approach: black-box supervised (Bb-S), grey-box supervised (Gb-S), and white-box supervised (Wb-S). These models have the same pipeline of training the uncertainty estimation model and the difference is only on the availability of the LLM. For the Bb-S method, we use the Gemma-7B as the model $q_\theta$ to evaluate the uncertainty of LLaMA2-7B/LLaMA3-8B $p_\theta$ (treated as a black-box), and reversely, use LLaMA2-7B to evaluate Gemma-7B. The supervised uncertainty model $\hat{g}$ is trained based on the random forest model (Breiman, 2001). Details on the feature construction and the training of the random forest model are deferred to Appendix D.2.

**Performance metrics.** For the model evaluation, we follow Filos et al. (2019); Kuhn et al. (2023) and compare the performance of our methods against the benchmark using the generated uncertainty score to predict whether the answer is correct. The area under the receiver operator characteristic curve (AUROC) metric is employed to measure the performance of the uncertainty estimation. As noted in Section 4.1, AUROC works as a good metric for uncertainty estimation whereas for uncertainty calibration, we follow the more standard calibration metrics and present the results in Section C.

### 5.2 PERFORMANCE OF UNCERTAINTY ESTIMATION

#### 5.2.1 QUESTION ANSWERING AND MACHINE TRANSLATION

The question answering and machine translation tasks can all be viewed as natural language generation tasks so we present their results together. Table 1 summarizes the three versions of our proposed supervised method against the existing benchmarks in terms of AUROC.

Table 1: Out-of-sample AUROC performance for benchmarks and our methods on natural language generation tasks.

| Dataset | LLM | Benchmarks | | | | | | Ours | | |
| --- | --- | --- | --- | --- | --- | --- | --- | --- | --- | --- |
| | | MaxL | AvgL | MaxE | AvgE | SU | A4C | Bb-S | Gb-S | Wb-S |
| TriviaQA | G-7B | 0.857 | 0.862 | 0.849 | 0.854 | 0.847 | 0.534 | 0.879 | 0.866 | **0.882** |
| | L-7B | 0.565 | 0.761 | 0.761 | 0.773 | 0.678 | 0.526 | **0.925** | 0.811 | 0.897 |
| | L-8B | 0.838 | 0.851 | 0.849 | 0.853 | 0.826 | 0.571 | 0.843 | 0.861 | **0.874** |
| CoQA | G-7B | 0.710 | 0.708 | 0.725 | 0.708 | 0.674 | 0.515 | 0.737 | 0.737 | **0.762** |
| | L-7B | 0.535 | 0.600 | 0.603 | 0.580 | 0.541 | 0.502 | **0.848** | 0.667 | 0.807 |
| | L-8B | 0.692 | 0.697 | 0.716 | 0.699 | 0.684 | 0.506 | 0.745 | 0.737 | **0.769** |
| WMT-14 | G-7B | 0.668 | 0.589 | 0.637 | 0.811 | 0.572 | 0.596 | **0.863** | 0.829 | 0.855 |
| | L-7B | 0.606 | 0.712 | 0.583 | 0.711 | 0.513 | 0.506 | **0.792** | 0.724 | 0.779 |
| | L-8B | 0.554 | 0.685 | 0.616 | 0.729 | 0.510 | 0.502 | 0.700 | 0.724 | **0.745** |

We make several remarks on the numerical results. First, our methods generally have a better performance than the existing benchmarks. Note that the existing benchmarks are mainly unsupervised and based on one single score, and also that our method proceeds with the most standard pipeline for supervised training of an uncertainty estimation model. The advantage of our method should be attributed to the supervised nature and the labeled dataset. While these unsupervised benchmark methods can work in a larger scope than these NLP tasks (though they have not been extensively tested on open questions yet), our methods rely on the labeled dataset. But in addition to these better numbers, the experiment results show the potential of labeled datasets for understanding the uncertainty in LLM's responses. In particular, our method Gb-S uses features including the benchmark methods, and it shows that some minor supervised training can improve a lot upon the ad-hoc uncertainty estimation based on one single score such as MaxL or MaxE.

Second, our method Wb-S has a clear advantage over our method Gb-S. Note that these two methods differ in that the Wb-S uses the hidden activations while the Gb-S only uses probability-related (and entropy-related) features. This implies that the hidden activations do contain uncertainty information which we will investigate more in Appendix B. Also, we note from the table that there is no single unsupervised grey-box method (under the Benchmarks columns) that consistently surpasses others across different datasets/NLP tasks. For example, among all these unsupervised benchmark methods for grey-box LLMs, AvgE emerges as a top-performing one for the Gemma-7B model when applied

to the machine translation task, but it shows the poorest performance for the same Gemma-7B model when tested on the question-answering CoQA dataset. This inconsistency highlights some caveats when using the unsupervised approach for uncertainty estimation of LLMs.

Lastly, we note that the Bb-S method has a similar or even better performance as the Wb-S method. As discussed in Section 3.3, the performance of uncertainty estimation relies on the LLM that we use to evaluate the prompt-response pair. Therefore, it is not surprising to see that in the question-answering task, for answers generated by LLaMA2-7B, Bb-S features better uncertainty estimation than Wb-S, possibly because Gemma-7B, the LLM that is used as the "tool LLM" in Algorithm 1, encodes better knowledge about the uncertainty of the answers than LLaMA-7B. We also note that the performance of Bb-S is not always as good as Wb-S, and we hypothesize that it is because LLMs' output distribution differs, which could result in evaluating the uncertainty of different answers. Despite these inconsistencies, the performance of Bb-S is still strong, and these results point to a potential future avenue for estimating the uncertainty of closed-source LLMs.

### 5.2.2 MULTIPLE CHOICE (MMLU)

Table 2 presents the performance of our methods against the benchmark methods on the MMLU dataset. For this multiple choice task, the output is from {A,B,C,D} which bears no semantic meaning, and therefore we do not include the Semantic Uncertainty (SU) as Table 1. The results show the advantage of our proposed supervised approach, consistent with the previous findings in Table 1.

Table 2: Out-of-sample AUROC performance for benchmarks and our methods on the MMLU dataset. The column Probability represents using the probability of the most likely choice as the uncertainty metric. The column Entropy represents the entropy of the distribution over the choices.

| Model | Benchmarks | | | Ours | | |
|---|---|---|---|---|---|---|
| | Probability | Entropy | A4C | Bb-S | Gb-S | Wb-S |
| Gemma-7B | 0.712 | 0.742 | 0.582 | 0.765 | 0.776 | **0.833** |
| LLaMA2-7B | 0.698 | 0.693 | 0.514 | **0.732** | 0.698 | 0.719 |
| LLaMA3-8B | 0.781 | 0.791 | 0.516 | 0.766 | 0.793 | **0.830** |

We defer more numerical experiments and visualization to Appendices B and C where we investigate more on (i) the effect of the choice of layers; (ii) the scale of the LLMs used; (iii) the *uncertainty neurons* of the LLMs; and (iv) the calibration performance.

### 5.3 TRANSFERABILITY

In this subsection, we evaluate the robustness of our methods under the OOD setting.

**Setup for the OOD multiple-choice task.** We split the MMLU datasets into two groups based on the subjects: Group 1 contains questions from the first 40 subjects while Group 2 contains the remaining 17 subjects, such that the test dataset size of each group is similar (around 600 questions). Note that these 57 subjects span a diverse range of topics, and this means the training and test set can be very different. To test the OOD robustness, we train the proposed methods on one group and evaluate the performance on the other group.

**Setup for the OOD question-answering task.** For the QA task, since we have two datasets (CoQA and TriviaQA), we train the supervised model on either the TriviaQA or CoQA dataset and then evaluate its performance on the other dataset. While both datasets are for question-answering purposes, they diverge notably in two key aspects: (i) CoQA prioritizes assessing the LLM's comprehension through the discernment of correct responses within extensive contextual passages, while TriviaQA focuses on evaluating the model's recall of factual knowledge. (ii) TriviaQA typically contains answers comprising single words or short phrases, while CoQA includes responses of varying lengths, ranging from shorter to more extensive answers.

Table 3 summarizes the performance of these OOD experiments. As expected, for all the methods, there is a slight drop in terms of performance compared to the in-distribution setting (reported by the numbers in the parentheses in the table). We make the following observations based on the experiment results. First, based on the performance gap between in-distribution and OOD evalua-

Table 3: Transferability of the trained uncertainty estimation model across different groups of subjects in MMLU and question-answering datasets. For our proposed Bb-S, Gb-S, and Wb-S methods, values within the parentheses ($\cdot$) represent the AUROCs where the uncertainty estimation model is trained and tested on the same group of subjects or dataset, while values outside the parentheses represent models trained on another group of subjects or dataset. The Best GB and Best BB columns refer to the best AUROC achieved by the unsupervised grey-box benchmarks and black-box benchmarks (fully listed in Table 1 and Table 2), respectively.

| LLMs | Test data | Ours | | | Best of benchmarks | |
| | | Bb-S | Gb-S | Wb-S | Best GB | Best BB |
|---|---|---|---|---|---|---|
| colspan Transferability in MMLU | | | | | | |
| G-7B | Group 1 | 0.756(0.768) | 0.793(0.799) | 0.846(0.854) | 0.765 | 0.538 |
| | Group 2 | 0.738(0.760) | 0.755(0.754) | 0.804(0.807) | 0.721 | 0.616 |
| L-7B | Group 1 | 0.733(0.749) | 0.715(0.713) | 0.726(0.751) | 0.719 | 0.504 |
| | Group 2 | 0.700(0.714) | 0.676(0.677) | 0.685(0.692) | 0.679 | 0.529 |
| L-8B | Group 1 | 0.763(0.773) | 0.796(0.795) | 0.836(0.839) | 0.799 | 0.524 |
| | Group 2 | 0.729(0.761) | 0.786(0.785) | 0.794(0.818) | 0.782 | 0.507 |
| colspan Transferability in Question-Answering Datasets | | | | | | |
| G-7B | TriviaQA | 0.842(0.879) | 0.861(0.866) | 0.861(0.882) | 0.862 | 0.847 |
| | CoQA | 0.702(0.737) | 0.722(0.737) | 0.730(0.762) | 0.725 | 0.674 |
| L-7B | TriviaQA | 0.917(0.925) | 0.801(0.811) | 0.881(0.897) | 0.773 | 0.678 |
| | CoQA | 0.825(0.848) | 0.623(0.667) | 0.764(0.807) | 0.603 | 0.541 |
| L-8B | TriviaQA | 0.813(0.843) | 0.859(0.861) | 0.863(0.874) | 0.853 | 0.826 |
| | CoQA | 0.710(0.745) | 0.714(0.737) | 0.725(0.769) | 0.716 | 0.684 |

tion, it is evident that although incorporating white-box features such as hidden activations makes the model more susceptible to performance decreases on OOD tasks, these features also enhance the uncertainty estimation model's overall capacity, and the benefits outweigh the drawbacks. It is also noteworthy that even in these scenarios of OOD, our Wb-S and Bb-S method almost consistently outperform corresponding benchmarks. Overall, the robustness of our methods shows that the hidden layers' activations within the LLM exhibit similar patterns in encoding uncertainty information to some extent. The performance drop (from in-distribution to OOD) observed in the MMLU dataset is notably less than that in the question-answering dataset, which may stem from the larger disparity between the CoQA and TriviaQA datasets compared to that between two distinct groups of subjects within the same MMLU dataset. This suggests that in cases of significant distributional shifts, re-training or re-calibrating the uncertainty estimation model using test data may be helpful.

## 6 CONCLUSIONS

In this paper, we study the problem of uncertainty estimation and calibration for LLMs. We follow a simple and standard supervised idea and use the labeled NLP datasets to train an uncertainty estimation model for LLMs. Our finding is that, first, the proposed supervised methods have better performances than the existing unsupervised methods. Second, the hidden activations of the LLMs contain uncertainty information about the LLMs' responses. Third, the black-box regime of our approach (Bb-S) provides a new approach to estimating the uncertainty of closed-source LLMs. Lastly, we distinguish the task of uncertainty estimation from uncertainty calibration and show that a better uncertainty estimation model leads to better calibration performance. One limitation of our proposed supervised method is that it critically relies on the labeled data. For the scope of our paper, we restrict the discussion to the NLP tasks and datasets. One future direction is to utilize the human-annotated data for LLMs' responses to train a supervised uncertainty estimation model for open-question prompts. We believe the findings that the supervised method gives a better performance and the hidden activations contain the uncertainty information will persist.

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

## A  MORE RELATED LITERATURE

**Hallucination detection.**  Recently, there is a trend of adopting uncertainty estimation approaches for hallucination detection. The rationale is that the information of the value of logits and the hidden states contain some of the LLMs' beliefs about the trustworthiness of its generated output. By taking the activations of hidden layers as input, Azaria & Mitchell (2023) train a classifier to predict hallucinations, and Verma et al. (2023) develop epistemic neural networks aimed at reducing hallucinations. Slobodkin et al. (2023) demonstrate that the information from hidden layers of LLMs' output can indicate the answerability of an input query, providing indirect insights into hallucination occurrences. Chen et al. (2024) develop an unsupervised metric that leverages the internal states of LLMs to perform hallucination detection. More related works on hallucination detection can be found in CH-Wang et al. (2023); Duan et al. (2024); Xu et al. (2024). While there is a lack of a rigorous definition of hallucination, and its definition varies in the above-mentioned literature, the uncertainty estimation problem can be well defined, and our results on uncertainty estimation can also help the task of hallucination detection.

**Leveraging LLMs' hidden activation.** The exploration of hidden states within LLMs has been studied to better understand LLMs' behavior. Mielke et al. (2022) improve the linguistic calibration performance of a controllable chit-chat model by fine-tuning it using a calibrator trained on the hidden states, Burns et al. (2022) utilizes hidden activations in an unsupervised way to represent knowledge about the trustfulness of their outputs. Liu et al. (2023) show that LLMs' linguistic outputs and their internal states can offer conflicting information about truthfulness, and determining whether outputs or internal states are more reliable sources of information often varies from one scenario to another. By taking the activations of hidden layers as input, Ahdritz et al. (2024) employ a linear probe to show that hidden layers' information from LLMs can be used to differentiate between epistemic and aleatoric uncertainty. Duan et al. (2024) experimentally reveal the variations in hidden layers' activations when LLMs generate true versus false responses in their hallucination detection task. Lastly, Li et al. (2024) enhance the truthfulness of LLMs during inference time by adjusting the hidden activations' values in specific directions.

We also remark on the following two aspects:

- Fine-tuning: For all the numerical experiments in this paper, we do not perform any fine-tuning with respect to the underlying LLMs. While the fine-tuning procedure generally boosts the LLMs' performance on a downstream task, our methods can still be applied for a fine-tuned LLM, which we leave as future work.

- Hallucination: The hallucination problem has been widely studied in the LLM literature. Yet, as mentioned earlier, it seems there is no consensus on a rigorous definition of what hallucination refers to in the context of LLMs. For example, when an image classifier wrongly classifies a cat image as a dog, we do not say the image classifier hallucinates, then why or when we should say the LLMs hallucinate when they make a mistake? Comparatively, the uncertainty estimation problem is more well-defined, and we provide a mathematical formulation for the uncertainty estimation task for LLMs. Also, we believe our results on uncertainty estimation can also help with a better understanding of the hallucination phenomenon and tasks such as hallucination detection.

## B INTERPRETING THE UNCERTAINTY ESTIMATION

Now we use some visualizations to provide insights into the working mechanism of the uncertainty estimation procedure for LLMs and to better understand the experiment results in the previous subsection.

### B.1 LAYER COMPARISON

For general LLMs, each token is associated with a relatively large number of hidden layers (32 layers for LLaMA2-7B for example), each of which is represented by high-dimensional vectors (4096 for LLaMA2-7B). Thus it is generally not a good practice to incorporate all hidden layers as features for the uncertainty estimation due to this dimensionality. Previous works find that the middle layer and the last layer activations of the LLM's last token contain the most useful features for supervised learning (Burns et al., 2022; Chen et al., 2024; Ahdritz et al., 2024; Azaria & Mitchell, 2023). To investigate the layer-wise effect for uncertainty estimation, we implement our Wb-S method with features different in two aspects: (i) different layers within the LLM architecture, specifically focusing on the middle and last layers (e.g., LLaMA2-7B and LLaMA3-8B: 16th and 32nd layers out of 32 layers with 4096 dimensions; Gemma-7B: 14th and 28th layers out of 28 layers with 3072 dimensions); and (ii) position of token activations, including averaging hidden activations over all the prompt/answer tokens or utilizing the hidden activation of the last token. The second aspect makes sense when the output contains more than one token, so we conduct this experiment on the natural language generation tasks only. Figure 3 gives a visualization of the comparison result. While the performances of these different feature extraction ways are quite similar in terms of performance across different tasks and LLMs, activation features from the middle layer generally perform better than the last layer. This may come from the fact that the last layer focuses more on the generation of the next token instead of summarizing information of the whole sentence, as has been discussed by Azaria & Mitchell (2023).

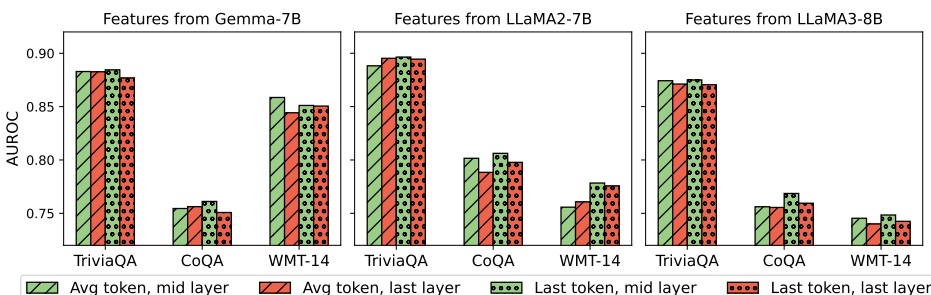

Figure 3: Performance comparison of using hidden activations from different tokens and layers as features in the Wb-S method. The bars filled with '/' and '.' represent the activations averaged over the answer tokens and the hidden activation of the last token, respectively. And the green and orange bars denote the activations from the middle and the last layer, respectively.

## B.2 Scaling effect

In Figure 4, we investigate whether hidden activations from larger LLMs enhance our uncertainty estimation method. For a fair comparison, we fix the target LLM that generates the output in Algorithm 1 and vary the tool LLM used for analysis. For example, in the left plot of Figure 4, we use Gemma-7B to generate the outputs, and LLaMA2-7B, LLaMA2-13B, and Gemma-7B to perform uncertainty estimation.

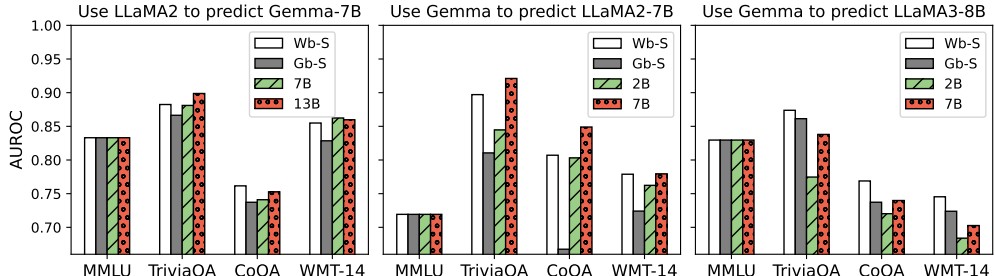

Figure 4: (Left) Using the hidden activations of LLaMA2-7B and LLaMA2-13B to estimate the uncertainty of the answer provided by Gemma-7B. (Middle) Using the hidden activations of Gemma-2B and Gemma-7B to estimate the uncertainty of the answer provided by LLaMA2-7B. (Right) Using the hidden activations of Gemma-2B and Gemma-7B to estimate the uncertainty of the answer provided by LLaMA3-8B

We find that larger LLM does encode better knowledge about the uncertainty, which is attributed to their improved knowledge in answering the questions. We also note that in the case of using Gemma to predict LLaMA2-7B, even a small tool LLM (Gemma-2B) is capable of achieving better performance than the Gb-S that only uses the entropy- and probability-related features from the target LLM. This result also underscores the benefits of adopting the internal state in estimating the uncertainty, even from an LLM different from the one generating the answers.

## B.3 Histogram of correlations

Figure 5 plots the histograms of the pairwise correlations between the neuron activations and the labels (whether the LLM's response is correct). We make two observations here: First, for both LLMs, some neurons have a significantly positive (or negative) correlation with the label. We can interpret these neurons as the *uncertainty neuron* for the corresponding task. When these neurons are activated, the LLMs are uncertain about their responses. Second, Gemma-7B and LLaMA3-8B have more significant neurons than LLaMA2-7B, and this is consistent with the better performance of Gemma-7B and LLaMA3-8B in Table 1 and Table 2. Also, this reinforces that the hidden activations of the LLMs contain uncertainty information about the LLM's output.

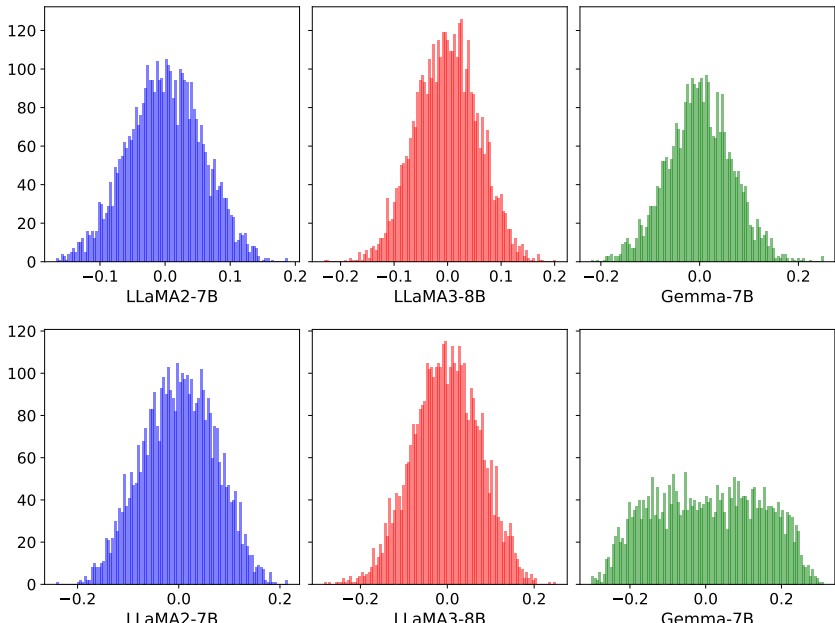

Figure 5: The histograms of the pairwise correlations on the TriviaQA task between the neuron activations and the labels (whether the LLM's response is correct), where the neural values are the last-token hidden activations of answers from the middle layer (upper) and the last layer (lower) of two models respectively.

Figure 6 plots some example neurons' activation by selecting the neurons with the largest absolute correlations in Figure 5. More neurons from the last layer can be found in Figure 7. These neurons as an individual indicator exhibit different distributional patterns when the response is correct compared to when the response is incorrect, and thus reflect the uncertainty of the LLM's responses.

### B.4 PROOF OF PROPOSITION 4.1

The proof of Proposition 4.1 follows from the definition of $f^*$.

## C CALIBRATION PERFORMANCE

In Section 4.1, we distinguish the two tasks of uncertainty estimation and uncertainty calibration. Throughout the paper, we have been focused on improving the performance on the task of uncertainty estimation – to predict when the LLM is uncertain about its response. Generally, a better uncertainty estimation model leads to one with better calibration performance. The calibration (or recalibration) of the uncertainty estimation model can be indeed reduced to the classic ML setting which does not involve the LLM. Table 4 gives the calibration performance and we see an advantage of our supervised methods over benchmark methods consistent with the AUROC performance in Table 1. We adopt the histogram binning method here because we find that the temperature scaling method and the Platt scaling method will give all predicted scores concentrated within a small range such as $[0.2, 0.6]$. We also do not exclude the possibility that the other calibration methods can give even better performance. The point to make here is that uncertainty estimation and uncertainty calibration are two closely related tasks. Note that (i) a better uncertainty estimation model leads to a better calibration performance and (ii) the LLMs are pretrained and not designed for these NLP tasks in the first place (see Section 4.2) so that there is no uncertainty score readily available (as the predicted probabilities for the image classifiers); we emphasize the importance of an extra uncertainty estimation procedure as our supervised one so to extract the uncertainty information from the inside of the LLMs.

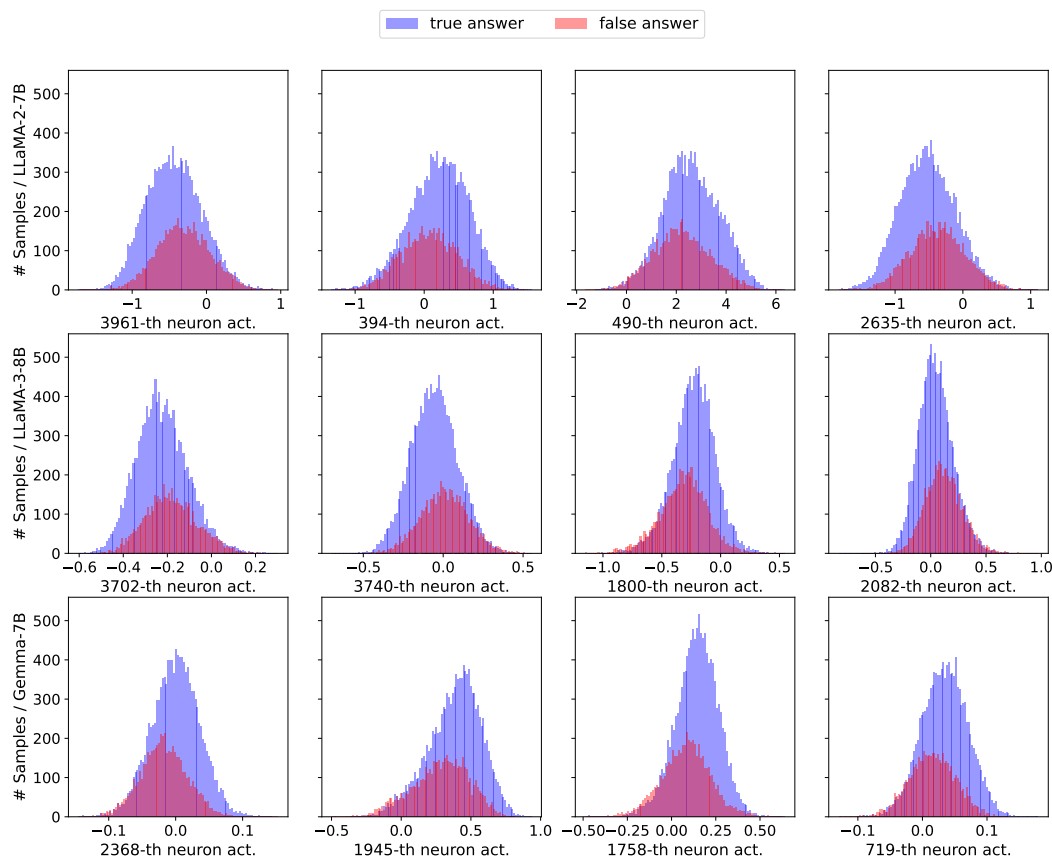

Figure 6: Distribution of values from particular neurons of mid-layers on TriviaQA dataset.

## D DETAILS FOR THE NUMERICAL EXPERIMENTS

We ran all of our experiments on an AMD EPYC 7452 128-core processor with 4×48G NVIDIA A6000 GPUs.

### D.1 DATASET PREPARATION

In the following we provide more information for the three tasks considered in our numerical experiments.

- Question answering. We follow Kuhn et al. (2023) and use the CoQA and TriviaQA (Joshi et al., 2017) datasets. The CoQA task requires the LLM to answer questions by understanding the provided text, and the TriviaQA requires the LLM to answer questions based on its pre-training knowledge. We adopt the scoring function $s(\cdot, \cdot)$ as Rouge-1 (Lin & Och, 2004b) and label a response $\boldsymbol{y}_i$ as correct if $s(\boldsymbol{y}_i, \boldsymbol{y}_{i,\text{true}}) \geq 0.3$ and incorrect otherwise.

- Multiple choice. We consider the Massive Multitask Language Understanding (MMLU) dataset (Hendrycks et al., 2020), a collection of 15,858 questions covering 57 subjects across STEM. Due to the special structure of the dataset, the generated output $\boldsymbol{y}_i$ and the correct answer $\boldsymbol{y}_{\text{true},i} \in \{A, B, C, D\}$. Therefore, this task can also be regarded as a classification problem for the LLM by answering the question with one of the four candidate choices.

- Machine translation. We consider the WMT 2014 dataset (Bojar et al., 2014) for estimating LLM's uncertainty on the machine translation task. The scoring function $s(\cdot, \cdot)$ is chosen to be the BLEU score (Papineni et al., 2002; Lin & Och, 2004a) and the generated answer $\boldsymbol{y}_i$ is labeled as correct if $s(\boldsymbol{y}_i, \boldsymbol{y}_{i,\text{true}}) > 0.3$ and incorrect otherwise.

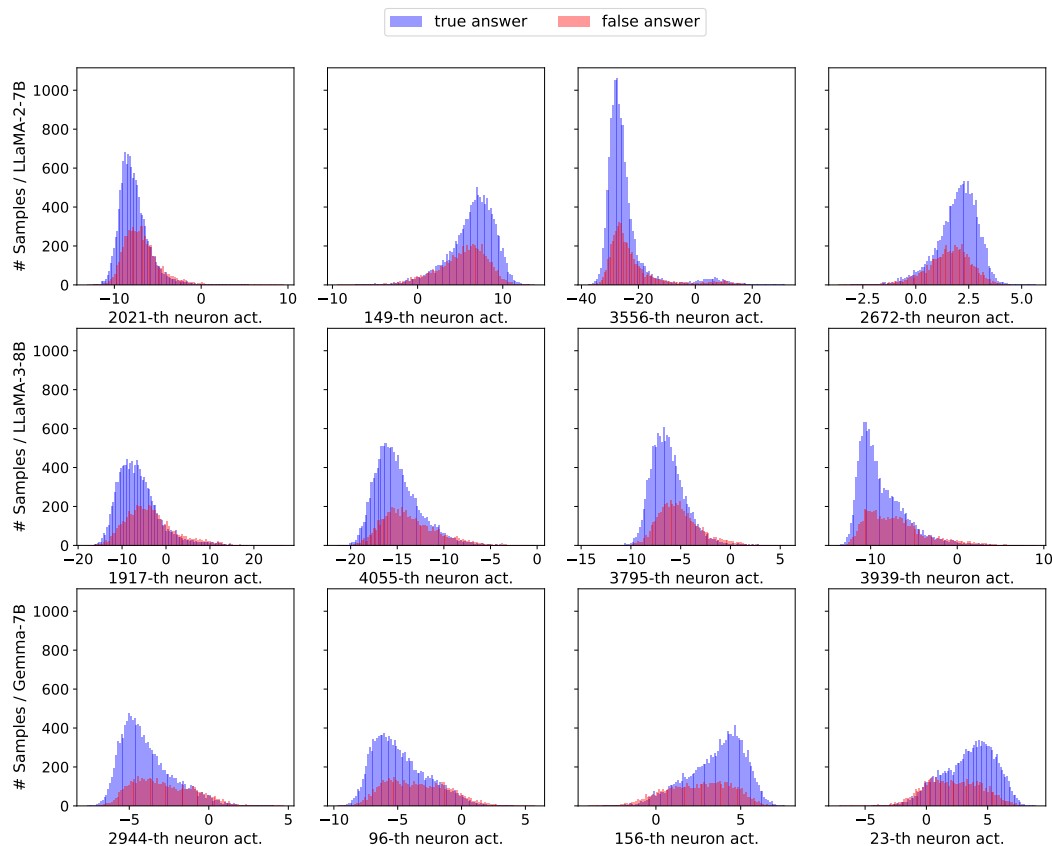

Figure 7: More distribution of values from specific neurons of last layers on the TriviaQA dataset. The plots are obtained in the same way as Figure 6.

Table 4: Calibration performance on natural language generation tasks after histogram binning. The base models are from Table 1. The original uncertainty scores from the base models are first scaled into $[0, 1]$ and then a histogram binning is performed with 20 bins of equal length.

| Metric | Dataset | Model | Benchmarks | | | | | | Ours | | |
| --- | --- | --- | --- | --- | --- | --- | --- | --- | --- | --- | --- |
| | | | MaxL | AvgL | MaxE | AvgE | SU | A4C | Bb-S | Gb-S | Wb-S |
| NLL | TriviaQA | G-7B | 0.478 | 0.500 | 0.428 | 0.472 | 0.739 | 8.710 | 0.414 | 0.467 | 0.392 |
| | | L-7B | 1.155 | 0.551 | 0.575 | 0.600 | 1.481 | 21.119 | 0.338 | 0.580 | 0.388 |
| | | L-8B | 0.483 | 0.407 | 0.383 | 0.401 | 0.719 | 8.515 | 0.423 | 0.467 | 0.365 |
| | CoQA | G-7B | 0.778 | 0.474 | 0.469 | 0.476 | 0.632 | 8.106 | 0.474 | 0.497 | 0.457 |
| | | L-7B | 1.047 | 0.620 | 0.637 | 0.649 | 1.358 | 11.708 | 0.417 | 0.607 | 0.457 |
| | | L-8B | 0.823 | 0.502 | 0.508 | 0.499 | 0.762 | 8.007 | 0.551 | 0.535 | 0.507 |
| | WMT-14 | G-7B | 9.674 | 1.266 | 0.809 | 0.618 | 0.701 | 17.933 | 0.454 | 0.463 | 0.449 |
| | | L-7B | 1.204 | 1.150 | 0.718 | 0.809 | 0.796 | 16.913 | 0.553 | 0.622 | 0.583 |
| | | L-8B | 1.490 | 0.752 | 0.652 | 0.676 | 0.722 | 21.340 | 0.649 | 0.673 | 0.612 |
| ECE | TriviaQA | G-7B | 0.152 | 0.138 | 0.066 | 0.115 | 0.275 | 0.253 | 0.056 | 0.075 | 0.067 |
| | | L-7B | 0.437 | 0.068 | 0.048 | 0.146 | 0.188 | 0.616 | 0.043 | 0.087 | 0.049 |
| | | L-8B | 0.171 | 0.082 | 0.046 | 0.081 | 0.196 | 0.283 | 0.107 | 0.087 | 0.075 |
| | CoQA | G-7B | 0.356 | 0.054 | 0.112 | 0.064 | 0.221 | 0.237 | 0.121 | 0.129 | 0.113 |
| | | L-7B | 0.397 | 0.065 | 0.105 | 0.073 | 0.174 | 0.494 | 0.052 | 0.071 | 0.038 |
| | | L-8B | 0.339 | 0.031 | 0.071 | 0.033 | 0.196 | 0.312 | 0.156 | 0.110 | 0.122 |
| | WMT-14 | G-7B | 0.499 | 0.464 | 0.234 | 0.197 | 0.072 | 0.521 | 0.097 | 0.063 | 0.073 |
| | | L-7B | 0.164 | 0.389 | 0.065 | 0.269 | 0.127 | 0.491 | 0.045 | 0.090 | 0.101 |
| | | L-8B | 0.318 | 0.192 | 0.051 | 0.142 | 0.029 | 0.618 | 0.145 | 0.201 | 0.137 |
| Brier | TriviaQA | G-7B | 0.282 | 0.221 | 0.224 | 0.215 | 0.344 | 0.279 | 0.266 | 0.288 | 0.282 |
| | | L-7B | 0.431 | 0.241 | 0.271 | 0.259 | 0.322 | 0.645 | 0.334 | 0.322 | 0.315 |
| | | L-8B | 0.262 | 0.192 | 0.204 | 0.188 | 0.291 | 0.373 | 0.258 | 0.265 | 0.255 |
| | CoQA | G-7B | 0.318 | 0.174 | 0.188 | 0.171 | 0.232 | 0.241 | 0.207 | 0.218 | 0.212 |
| | | L-7B | 0.395 | 0.233 | 0.242 | 0.230 | 0.265 | 0.464 | 0.296 | 0.256 | 0.276 |
| | | L-8B | 0.338 | 0.197 | 0.201 | 0.191 | 0.255 | 0.359 | 0.258 | 0.242 | 0.248 |
| | WMT-14 | G-7B | 0.505 | 0.454 | 0.330 | 0.319 | 0.247 | 0.606 | 0.327 | 0.287 | 0.309 |
| | | L-7B | 0.313 | 0.413 | 0.271 | 0.334 | 0.275 | 0.502 | 0.296 | 0.277 | 0.288 |
| | | L-8B | 0.343 | 0.279 | 0.250 | 0.263 | 0.246 | 0.620 | 0.282 | 0.300 | 0.284 |

**Prompt dataset generation.** For all the tasks studied in this paper, we adopt the few-shot prompting for the LLM. Specifically, in the prompt, we provide $r$ examples to make the LLM learn the format of the response, as illustrated in the following. For the question-answering task, we construct the prompt without using any question-answering sample repeatedly in the original dataset. For example, Prompt 1 includes the 1st to $r$-th question-answering samples in the original dataset as the examples and the $(r+1)$-th sample as the target question-answering pair for the LLM; next, Prompt 2 uses the $(r+2)$-th to $(2r+1)$-th samples as the examples and the $(2r+2)$-th sample as the target question-answering pair. However, as the test datasets of MMLU and WMT used for evaluation are not sufficiently large, we generate the prompt in a convolution-like manner: Prompt 2 includes the 2nd to $(r+1)$-th question-answering samples as the examples and the $(r+2)$-th sample as the target question-answering pair.

**Dataset split.** After generating the prompt-answering dataset, we split this dataset into two parts for training the calibration model and evaluation/test. For the MMLU and WMT datasets, we take the dataset generated from the original validation/test dataset. For the question-answering task, as the answer of TriviaQA in the original test dataset is vacant, we take the first 2000 generated prompt-answering pairs from the training dataset as the test dataset, and the remaining for training.

**Prompting format**. Here we give the different prompting templates used for different tasks. We use few-shot prompting and the templates can always be roughly divided into four parts: introduction (empty only for WMT), examples, question, and answer, where examples are just $r$ distinct question-answer pairs in the same form as the question and answer parts. We feed the model with the template string except for the reference answer as inputs.

### D.2 DETAILS OF THE TRAINING PROCEDURE

For the three regimes of our supervised approach presented in Section 3.3, the details of the supervised training procedure are as below:

**Gb-S.** For the natural language generation tasks (question-answering and machine-translation), we train a random forest model with the input features listed in Table 5 (20 features in total). For the multiple-choice task, as the answer has only one token from {A, B, C, D}, we take the output logits of these 4 tokens (denoted as $\alpha_A$, $\alpha_B$, $\alpha_C$, and $\alpha_D$) after inputting the question prompt $x$ to the LLM. Then, we get the probability of each choice as follows:

$$p_\theta(y|\boldsymbol{x}) = \frac{\exp(\alpha_y)}{\sum_{y' \in \{A,B,C,D\}} \exp(\alpha_{y'})}, \ \forall y \in \{A, B, C, D\}.$$

Then we use 5 features as the input to Gb-S: the entropy of this distribution, and the sorted probability values in descending order.

**Wb-S.** The dimension of a hidden layer from LM is typically high (e.g., 4096 for LLaMA2-7B), which may prevent the calibration model from capturing the effective uncertainty information revealed from the activations, especially with limited training samples. Thus, before training a model, we do the feature selection first. We maintain all the features used in the Gb-S and select another 300 features (neural nodes): (i) We use all the features to train a Lasso model and select 100 neural nodes with the highest absolute coefficient values; (ii) By calculating the mutual information between any neural node and the label (correct or not), we select another 100 features possessing top absolute

Table 5: Grey-box features used for the supervised task of uncertainty estimation for LLMs.

| Name | Features from the response/answer | Features from the prompt/question |
|------|-----------------------------------|-----------------------------------|
| Max Ent | $\max_{j\in\{1,...,m\}} H(p_\theta(\cdot|\boldsymbol{x},\boldsymbol{y}_{1:j-1}))$ | $\max_{j\in\{1,...,n\}} H(p_\theta(\cdot|\boldsymbol{x}_{1:j-1}))$ |
| Min Ent | $\min_{j\in\{1,...,m\}} H(p_\theta(\cdot|\boldsymbol{x},\boldsymbol{y}_{1:j-1}))$ | $\min_{j\in\{1,...,n\}} H(p_\theta(\cdot|\boldsymbol{x}_{1:j-1}))$ |
| Avg Ent | $\frac{1}{m}\sum_{j=1}^{m} H(p_\theta(\cdot|\boldsymbol{x},\boldsymbol{y}_{1:j-1}))$ | $\frac{1}{n}\sum_{j=1}^{n} H(p_\theta(\cdot|\boldsymbol{x}_{1:j-1}))$ |
| Std Ent | $\sqrt{\frac{\sum_{j=1}^{m}(H(p_\theta(\cdot|\boldsymbol{x},\boldsymbol{y}_{1:j-1}))-\text{Avg Ent})^2}{m-1}}$ | $\sqrt{\frac{\sum_{j=1}^{n}(H(p_\theta(\cdot|\boldsymbol{x}_{1:j-1}))-\text{Avg Ent})^2}{n-1}}$ |
| Max Likelihood | $\max_{j\in\{1,...,m\}} -\log p_\theta(y_j|\boldsymbol{x},\boldsymbol{y}_{1:j-1})$ | $\max_{j\in\{1,...,n\}} -\log p_\theta(x_j|\boldsymbol{x}_{1:j-1})$ |
| Min Likelihood | $\min_{j\in\{1,...,m\}} -\log p_\theta(y_j|\boldsymbol{x},\boldsymbol{y}_{1:j-1})$ | $\min_{j\in\{1,...,n\}} -\log p_\theta(x_j|\boldsymbol{x}_{1:j-1})$ |
| Avg Likelihood | $\frac{1}{m}\sum_{j=1}^{m} -\log p_\theta(y_j|\boldsymbol{x},\boldsymbol{y}_{1:j-1})$ | $\frac{1}{n}\sum_{j=1}^{n} -\log p_\theta(x_j|\boldsymbol{x}_{1:j-1})$ |
| Std Likelihood | $\sqrt{\frac{\sum_{j=1}^{m}(-\log p_\theta(y_j|\boldsymbol{x},\boldsymbol{y}_{1:j-1})-\text{Avg Likelihood})^2}{m-1}}$ | $\sqrt{\frac{\sum_{j=1}^{n}(-\log p_\theta(x_j|\boldsymbol{x}_{1:j-1})-\text{Avg Likelihood})^2}{n-1}}$ |
| Avg Prob | $\frac{1}{m}\sum_{j=1}^{m} p_\theta(y_j|\boldsymbol{x},\boldsymbol{y}_{1:j-1})$ | $\frac{1}{n}\sum_{j=1}^{n} p_\theta(x_j|\boldsymbol{x}_{1:j-1})$ |
| Std Prob | $\sqrt{\frac{\sum_{j=1}^{m}(p_\theta(y_j|\boldsymbol{x},\boldsymbol{y}_{1:j-1})-\text{Avg Prob})^2}{m-1}}$ | $\sqrt{\frac{\sum_{j=1}^{n}(p_\theta(x_j|\boldsymbol{x}_{1:j-1})-\text{Avg Prob})^2}{n-1}}$ |

mutual information; (iii) We select another 100 features with top absolute Pearson correlation coefficient. After the feature selection, we train a random forest model to predict whether the response is correct based on the selected features.

In the experiment section of the main text, the features in the Wb-S for natural language generation tasks include (i) all the features used in the Gb-S, (ii) the hidden activations of the last token of the question from the middle layer (LLaMA2-7B or LLaMA3-8B: 16th layer; Gemma-7B: 14th layer), and (iii) the hidden activations of the last token of the answer from the middle layer. Therefore, in these natural language generation tasks, the dimension is 8212 for LLaMA2-7B/LLaMA3-8B and 6164 for Gemma-7B.

The features in the Wb-S for the multiple-choice task include (i) all the features used in the Gb-S and (ii) the hidden activations of the last token of the answer (letter A, B, C, or D) from the middle layer. The dimension is 4101 for LLaMA2-7B/LLaMA3-8B and 3077 for Gemma-7B.

Notably, there are many choices of the hidden activations employed in the Wb-S. Besides what has been shown in Section B, we provide further discussion in Section E.

**Bb-S.** The idea of building a supervised calibration model for a black-box LLM is to use the hidden layers and output distributions from another open-source LLM model by feeding it with the question and the provided response. Therefore, the features available for the Wb-S are also available for the open-source LLM, so we just take the corresponding features from the open-source LLM in the Bb-S. Hence, in the natural language generation tasks, the input dimension of the calibration model is 4196 (including hidden activations of the question and answer and 20 entropy and likelihood-related features, $2 \times 2048 + 20$) for Gemma-2B, 6164 for Gemma-7B, 8212 for LLaMA2-7B/LLaMA3-8B, and 10260 for LLaMA2-13B. In the multiple-choice task, the dimension is 2053 for Gemma-2B (including the hidden activations of the answer and 5 entropy- and probability-related features used in the Gb-S), 3077 for Gemma-7B, 4101 for LLaMA2-7B/LLaMA3-8B, and 5125 for LLaMA2-13B.

For all these methods, we employ the random forest (Breiman, 2001) using the implementation from the scikit-learn package (Pedregosa et al., 2011) to estimate the uncertainty. The hyperparameters are set as [n_estimators=150, random_state=0, max_depth=8, verbose=2, max_features=45] if the number of selected features is no less than 100 and [n_estimators=100, random_state=0, max_depth=4, verbose=2] otherwise.

# E    ADDITIONAL RESULTS AND VISUALIZATIONS

In Section B, we show the advantage of utilizing the hidden activations of the *answer* from the middle layer of the LLM to estimate the uncertainty in Wb-S. In this section, we further discuss the impact of employing the hidden activations from the *question* in the Wb-S.

The motivation stems from the observation that within the transformer architecture, although the hidden activation of a question's last token (referred to as the question's activation) is forwarded to obtain the hidden activation of the answer's last token (referred to as the answer's activation), implying that the answer's activation incorporates the question's activation information, it has been discovered that concatenating the question's activation with the answer's activation offers additional insights into the answer's uncertainty (Duan et al., 2024). We would like to further investigate the effectiveness of incorporating the question's activation along with the answer's activation into the supervised setting.

We experiment with three feature combinations in our supervised setting: (i) Question: we use the hidden activation of the last token of the question from the middle layer, incorporated with the entropy- or probability-related features of the question (10 features in total listed in the right column of Table 5) if it is a natural language generation task, otherwise incorporated with all the features in Gb-S; (ii) Answer: we use the hidden activation of the last token of the answer from the middle layer incorporated with all the features used in Gb-S; (iii) Question-Answer: we use the last-token hidden activation of both the question and answer from the middle layer and all the features in Gb-S. We compare their performance with Gb-S in Figure 8 and present the following observations.

**Question itself cannot capture enough uncertainty information.** From Figure 8, we observe that the method Bb-S consistently outperforms Question across all these tasks. This implies that incorporating the features relating to the question only cannot provide enough information about the uncertainty of the answer. This aligns with the inferior performance of the sample-based method (Kuhn et al., 2023) we tested in the earlier sections. In these methods, the uncertainty score is used to estimate the language model's uncertainty about the question. This result implies that uncertainty cannot be captured in the question by the language model without generating the answer.

**Question's hidden activation cannot help to generate more uncertainty information** Again from Figure 8, by comparing the performance of Answer and Question-Answer, we find that the inclusion of question's activation has little impact on improving the performance. This shows that the uncertainty from the question might have already been well encoded in the last token activation of the answer.

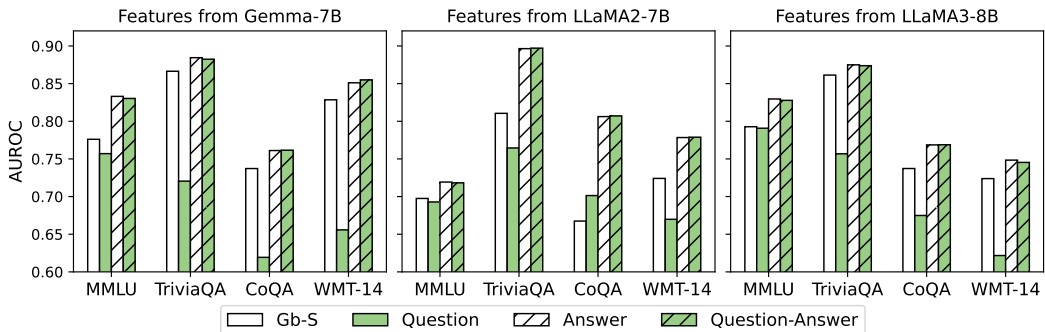

Figure 8: Performance comparison of using last-token middle layer hidden activations of the answer (Answer) or the concatenation of the question and answer (Question-Answer) as features in the Wb-S, where the features in Gb-S are also included in Wb-S. In the natural language generation tasks, the dimensions of Gb-S, Question, Answer, and Question-Answer for Gemma-7B are 20, 3082, 3092, and 6164, while for LLaMA2-7B or LLaMA3-8B they are 20, 4106, 4116, and 8212, respectively. In the MMLU task, for Gemma-7B they are 5, 3077, 3077, and 6149, while for LLaMA2-7B or LLaMA3-8B, they are 5, 4101, 4101, and 8197, respectively.

**The middle layer is still better than the last layer.** In Section B, Figure 3 shows that when using the hidden activation of the answer in the Wb-S, the middle layer of the LLM is a better choice than the last layer. The next question is: Does this conclusion still hold for using the concatenated hidden activations of the question and answer? We depict the experiment result in Figure 9, which is consistent with the conclusion drawn from Figure 3.

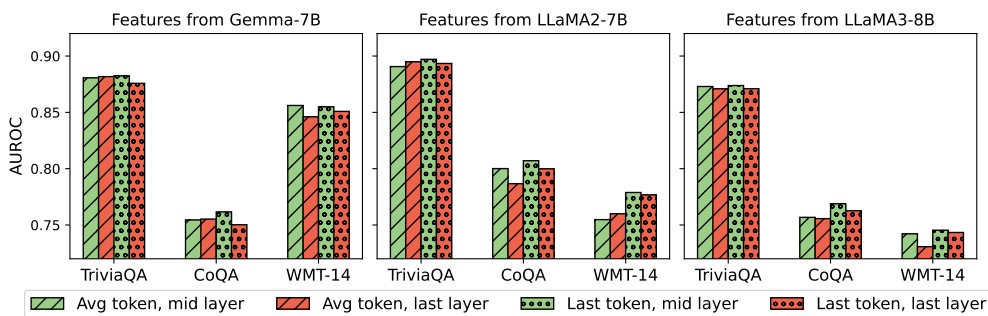

Figure 9: Performance comparison of using *question-answer concatenated* hidden activations from different tokens and layers as features in the Wb-S method. Scores are normalized in [0,1], where a lower value indicates larger uncertainty. For Gemma-7B, the dimension of the Wb-S input is 6164 (3072 from the question, 3072 from the answer, and 20 from the grey-box features). For LLaMA2-7B/LLaMA3-8B, it is 8212.

**Our method better characterizes the uncertainty.** We find that the grey-box and white-box features enhance the ability to characterize the dataset so that the distribution of the generated output's uncertainty score is better correlated with the output's correctness. According to Figure 10, we observe that with black-box features, the distributions of the uncertainty score for true and false answers are not very distinguishable, and the true answer's distribution is even similar to a uniform distribution. With grey-box and white-box features, the distributions of the uncertainty scores are more separated between the true and false answers. The results show the supervised learning approach not only achieves better AUROC but also learns to better separate the distribution of the uncertainty scores.

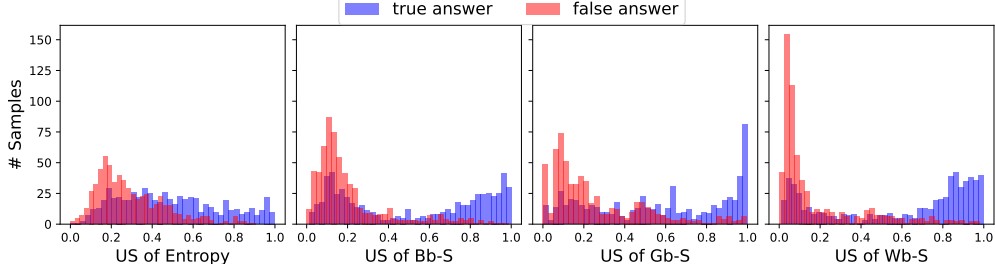

Figure 10: Uncertainty scores of different methods on the MMLU dataset for answers provided by the Gemma-7B model, where scores are normalized in [0,1], and US is short for uncertainty score. False answer refers to the sample where the choice assigned with maximum probability by the LLM is false, while true answer represents the sample answered correctly.

## F  EXAMPLES

In this section, we show some examples of the wrong answers the LLM generated and explore how different methods understand the LLM's uncertainty. The wrong answers are selected from those samples where the LLM makes wrong predictions.

Since we let the LLM output the greedy answer, which could be wrong, we expect an ideal uncertainty estimation model to output a high confidence score when the LLM generates the correct answer, and give a low confidence score when the LLM outputs the wrong answer. By looking at different wrong answers generated by the LLM, we note that although our approach sometimes gives a high confidence score on a wrong answer generated by the LLM, at other times it shows desirable properties such as giving higher uncertainty scores to better answers, and giving low confidence score when LLM does not know the answer.

Our illustrative examples are generated as follows: For questions where the LLM's greedy response is incorrect, we also extract the correct answer from the dataset and additional answers randomly generated by the LLM with lower probabilities than the greedy answer. Along with these answers, we also compute the answers' corresponding metrics and features so that we can observe how they behave with different outputs. We conduct this experiment in the test dataset of TriviaQA, in which both the question and answer are short. We summarize the ways that our uncertainty estimation model behaves as follows:

- **Confidently support a wrong answer.** The LLMs are confident that the wrong greedy answer is true and assign a high confidence score. Moreover, the LLMs give low uncertainty scores to the correct answers, suggesting a lack of knowledge about these questions. We give an example of LLaMA2-7B and Gemma-7B in Figure 11 and 12. Note that in both examples, our method assigns a low uncertainty score to the correct answer and a much higher uncertainty score to the wrong answer. In contrast, the unsupervised grey-box methods assign higher uncertainty scores to the correct answer.

- **Confidently reject a wrong answer.** We give examples from LLaMA2-7B and Gemma-7B in Figure 13 and 14. The uncertainty estimation model gives a higher score to the true answer or answers that are better than the wrong answer. This means that for these questions, our model actually knows which answer is better and can assign uncertainty scores accordingly. In contrast, the unsupervised methods tend to assign much higher uncertainty scores to the greedy (wrong) answer.

- **Unconfident about any answer.** Due to the lack of knowledge, the LLM may not know the true answer. We show the examples in Figure 15 and 16. From these examples, we can see that the model assigns almost the same uncertainty scores to these generated answers, including the true answer. In this scenario, the uncertainty estimation model is uncertain about the correctness of any answer. Furthermore, it is interesting to note that the unsupervised methods exhibit similar behavior, assigning almost similar scores to other answers as well, albeit with much higher uncertainty scores. This differs from the previous two cases, where the unsupervised method behaved differently from our uncertainty estimation model.

### An example of a confidently wrong answer (LM: LLaMA2-7B)

- **Question**: Who had a 70s No 1 hit with Billy, Don't Be A Hero?
- **Ref answer**: Bo Donaldson & The Heywoods

🤖 **Greedy answer**: Paper Lace

🤖 Answer 1: Bo Donaldson

🤖 Answer 2: Paperchaser

🤖 Answer 3: Paper Moon

| | Rouge-1 | Max Prob | Avg Prob | Max Ent | Avg Ent | Gb-S | Wb-S | Bb-S | SU | Ask4-conf |
|---|---|---|---|---|---|---|---|---|---|---|
| **Ref answer** | 1 | 0.13 | 0.94 | 0.82 | 0.94 | 0.21 | 0.31 | | | |
| **Greedy answer** | 0 | 0.79 | 0.99 | 0.86 | 0.94 | 0.82 | 0.83 | 0.72 | 0.31 | 0 |
| Answer 1 | 0.67 | 0.13 | 0.9 | 0.82 | 0.9 | 0.1 | 0.25 | | | |
| Answer 2 | 0 | 0 | 0.81 | 0.7 | 0.82 | 0.08 | 0.12 | | | |
| Answer 3 | 0 | 0 | 0.82 | 0.86 | 0.89 | 0.1 | 0.2 | | | |

Figure 11: An example of LLaMA2-7B assigning a confidently wrong answer in the TriviaQA dataset. Scores are normalized in $[0, 1]$, where a lower value indicates a larger uncertainty. The score of the greedy answer provided by any uncertainty estimation method is higher than that of the true answer, but the greedy answer is incorrect. The UK band Paper Lace did indeed release a version of "Billy, Don't Be A Hero" in 1974, the same year as the version of Bo, but it was Bo Donaldson & The Heywoods (a band in the U.S.) whose version topped the charts as a No.1 hit.

## An example of a confidently wrong answer (LM: Gemma-7B)

- **Question**: Which sitcom starred Leonard Rossiter in the role of a supermarket manager?
- **Ref answer**: Tripper's Day

🤖 **Greedy answer**: Rising Damp

🤖 Answer 1: Rising Damp.

🤖 Answer 2: The Rise and Fall of Reginald Perrin

| | Rouge-1 | Max Prob | Avg Prob | Max Ent | Avg Ent | Gb-S | Wb-S | Bb-S | SU | Ask4-conf |
|---|---|---|---|---|---|---|---|---|---|---|
| **Ref answer** | 1 | 0.00 | 0.66 | 0.70 | 0.74 | 0.14 | 0.15 | 0.24 | | |
| **Greedy answer** | 0 | 0.76 | 0.99 | 0.90 | 0.94 | 0.93 | 0.86 | 0.89 | 0.46 | 1 |
| Answer 1 | 0 | 0.02 | 0.87 | 0.81 | 0.88 | 0.60 | 0.40 | 0.86 | | |
| Answer 2 | 0 | 0.05 | 0.91 | 0.89 | 0.93 | 0.68 | 0.46 | 0.64 | | |

Figure 12: An example for Gemma-7B that assigns a high confidence score to a wrong answer. Leonard Rossiter starred in "Rising Damp" as a landlord, not as a supermarket manager.

## An example that the LM identifies the better answer (LM: LLaMA2-7B)

- **Question**: Which musical featured the songs A Secretary is Not A Toy, and The Company Way?
- **Ref answer**: How to Succeed in Business Without Really Trying

🤖 **Greedy answer**: The Pajama Game

🤖 Answer 1: How to Succeed In Business Without Really Trying

🤖 Answer 2: The Company Way

| | Rouge-1 | Max Prob | Avg Prob | Max Ent | Avg Ent | Gb-S | Wb-S | Bb-S | SU | Ask4-conf |
|---|---|---|---|---|---|---|---|---|---|---|
| **Ref answer** | 1 | 0.12 | 0.96 | 0.43 | 0.93 | 0.23 | 0.33 | | | |
| **Greedy answer** | 0 | 0.12 | 0.9 | 0.37 | 0.82 | 0.09 | 0.14 | 0.33 | 0.08 | 0 |
| Answer 1 | 1 | 0.08 | 0.93 | 0.43 | 0.94 | 0.14 | 0.22 | | | |
| Answer 2 | 0 | 0.01 | 0.78 | 0.37 | 0.6 | 0.08 | 0.13 | | | |

Figure 13: An example that LLaMA2-7B can successfully identify the better answer (by attaching a higher score). Scores are normalized in [0,1], where a lower value indicates larger uncertainty.

An example that the LM identifies the better answer (LM: Gemma-7B)

- **Question**: The behavior of sound in rooms and concert halls is a separate science. what is its name?
- **Ref answer**: Acoustics

**Greedy answer**: Acoustical

Answer 1: Acoustical Engineering

Answer 2: Acoustiics

| | Rouge-1 | Max Prob | Avg Prob | Max Ent | Avg Ent | Gb-S | Wb-S | Bb-S | SU | Ask4-conf |
|---|---|---|---|---|---|---|---|---|---|---|
| **Ref answer** | 1 | 0.45 | 0.96 | 0.86 | 0.88 | 0.64 | 0.73 | 0.93 | | |
| **Greedy answer** | 0 | 0.41 | 0.95 | 0.79 | 0.84 | 0.50 | 0.51 | 0.29 | 0.28 | 1 |
| Answer 1 | 0 | 0.28 | 0.94 | 0.79 | 0.83 | 0.39 | 0.44 | 0.33 | | |
| Answer 2 | 0 | 0.04 | 0.86 | 0.69 | 0.80 | 0.16 | 0.25 | 0.39 | | |

Figure 14: An example that Gemma-7B can successfully identify the better answer (by attaching a higher score). Scores are normalized in [0,1], where a lower value indicates larger uncertainty.

An example that the LM does not know the answer (LM: LLaMA2-7B)

- **Question**: Who played Sandy Richardson in the British tv series 'Crossroads'?
- **Ref answer**: Roger Tonge

**Greedy answer**: Noel Clarke

Answer 1: Mike Pratt

Answer 2: Lucy Carless

| | Rouge-1 | Max Prob | Avg Prob | Max Ent | Avg Ent | Gb-S | Wb-S | Bb-S | SU | Ask4-conf |
|---|---|---|---|---|---|---|---|---|---|---|
| **Ref answer** | 1 | 0.01 | 0.78 | 0.28 | 0.71 | 0.08 | 0.09 | | | |
| **Greedy answer** | 0 | 0.16 | 0.89 | 0.28 | 0.75 | 0.08 | 0.09 | 0.23 | 0 | 0 |
| Answer 1 | 0 | 0.01 | 0.82 | 0.28 | 0.73 | 0.08 | 0.09 | | | |
| Answer 2 | 0 | 0 | 0.71 | 0.28 | 0.63 | 0.08 | 0.08 | | | |

Figure 15: An example that LLaMA2-7B does not know the true answer. Scores are normalized in [0,1], where a lower value indicates larger uncertainty. The LM does not know the true answer and attempts to guess it by generating different names with low confidence scores, but the score is also low even when the LM faces the true answer.

## An example of the failure in estimating the uncertainty (LM: Gemma-7B)

- **Question**: What is the name of the colliery in the 1939 film 'The Stars Look Down'?
- **Ref answer**: Neptune Colliery

**Greedy answer**: The Black Diamond

Answer 1: Oakwood Colliery

Answer 2: Northmoor Colliery

|  | Rouge-1 | Max Prob | Avg Prob | Max Ent | Avg Ent | Gb-S | Wb-S | Bb-S | SU | Ask4-conf |
|---|---|---|---|---|---|---|---|---|---|---|
| **Ref answer** | 1 | 0 | 0.62 | 0.19 | 0.65 | 0.10 | 0.13 | 0.23 | | |
| **Greedy answer** | 0 | 0.02 | 0.72 | 0.18 | 0.20 | 0.10 | 0.10 | 0.12 | 0 | 1 |
| Answer 1 | 0 | 0 | 0.73 | 0.18 | 0.57 | 0.10 | 0.11 | 0.18 | | |
| Answer 2 | 0 | 0 | 0.73 | 0.18 | 0.53 | 0.10 | 0.12 | 0.19 | | |

Figure 16: An example that Gemma-7B does not know the true answer. Scores are normalized in [0,1], where a lower value indicates larger uncertainty.

