# OpenReview forum: "Uncertainty Estimation and Quantification for LLMs: A Simple Supervised Approach"
_ICLR.cc/2025/Conference — ICLR 2025 Conference Withdrawn Submission_

### Official Review · Reviewer_yeCh · 2024-10-22

**Soundness:** 2
**Presentation:** 3
**Contribution:** 2
**Rating:** 5
**Confidence:** 4

**Summary:**

Authors propose an uncertainty estimation method for LLMs that relies on a secondary LLM and a Random Forest to make estimates of the uncertainty for a given input and prediction. They show this setup gives better estimates than uncertainty estimation using the predictive entropy of predictions, verbalised uncertainty or semantic consistency between predictions for supervised NLP tasks, including some generalisation tasks.

**Strengths:**

- With the surge in popularity of LLMs, research in uncertainty estimation on LLMs is needed. As the authors discuss, this is distinct from uncertainty in traditional ML models.
- By providing a clear task definition for uncertainty estimation in Equation (1) authors give specific expectations. This is helpful for communication within the paper, but it also provides a clear context for future discussions.
- The proposed method appears promising and seems to offer substantially better uncertainty estimation under some circumstances.
- The idea that there may be information in the hidden activations in LLMs that is not in the final layer, while this wouldn’t be the case for other ML models is interesting and worth studying.

**Weaknesses:**

- The evaluation does not consider actual black/grey-box commercial LLMs such as ChatGPT, while this is (implicitly) what the authors propose their method for. If a black-box target LLM is substantially better than the tool LLM, the black-box based uncertainty estimates may outperform the features that rely on the tool-LLM activation. Results will be more persuasive if the experiment is applied to the most popular target LLMs.
- It may be expected that the uncertainty estimates of this method get substantially worse when the task is different from the supervised learning data, but the authors do not sufficiently explore this. An experiment where the uncertainty model is trained on Question-Answering but evaluated on Multiple-Choice might show when this model fails.
- Authors are effectively comparing a zero-shot uncertainty estimation method against a supervised-learning uncertainty estimation method. This is arguably not a fair comparison. Results would be more complete if authors additionally evaluate a “normal” supervised NLP model that does both the predictions and uncertainty estimates.
- The argument that there is extra information (w.r.t. to the output layer) in the hidden activations in LLMs that is otherwise not the case in normal ML models is interesting, but is missing a thorough investigation. Additionally, I think it has relatively limited relevance to the main portion of the work. I’d consider (if it aligns with the authors interest) to reserve the investigation into the hidden layers of an LLM for a separate dedicated paper, so this paper can maintain focused on one point. The evidence is currently fairly week that this information would exist. It may be that a model is not good at picking it out of the output layer, but that does not mean the information does not exist. The authors also don’t test a comparison in a normal NN. Overall, I think the discussion of hidden activations should not be considered as a core contribution or finding of the current paper.
- Throughout the paper there are no standard deviations reported for the results. Particularly when sometimes the differences between conditions are fairly small it is unclear whether differences are coincidental or significant.




## Additional Feedback (minor comments)
- It is unclear whether the AUROC scores in Tables 1 and 2 are for predicting the answer, or for predicting whether an answer is (in)correct. I assume these AUROC scores are for the uncertainty estimation, but it may be good to make this clearer.
- Table 3 should be made clearer by adding in bold the best performances. Additionally, to show whether whitebox features indeed suffer from more performance decrease than black-box features it would be good to add averages at the bottom of the table.
- On Line 38 it appears that the references (Gal & Ghahramani 20216, Lakshminarayanan et al. 2017, Guo et al. 2017, Minderer et al. 2021) claim that “the need for uncertainty estimation [is greater for] generative AI”, but this is not what those papers say. I understand the intention the authors have here (citing papers on UQ for normal ML), but with the current phrasing/placement it gives the wrong impression. Consider rephrasing.
- The (informal) definition of uncertainty estimation on Line 44 defines it as predicting quality, but then says that “quality” is confidence and uncertainty, which is circular reasoning. The “truthfulness” can be a good measure, but that cannot be the only measure of quality. I’d propose to reconsider the argument made here and rephrase.
- There’s a small language mistake in Line 90. “restrict its application to black-box LLMs” means “it can only be used on black-box LLMs”, but I think you mean the opposite.
- In Section 1.1 or Appendix A it might be appropriate to discuss verbalized uncertainty (such as the A4U method) to give a broad overview of existing work.
- Line 196 “The next is to …” -> “The next step is to …”
- There’s a typo in Line 213.
- In Tables 1 and 2 there’s a benchmark called “A4C”, but in the Benchmarks section it is defined as A4U.
- Appendix B.4 seems to be incomplete. The proof is not provided.

**Questions:**

The proposed task in Equation (1) g(x,y) may have some limitations. It is invariant to the model that generated the prediction y, making the uncertainty not necessarily an uncertainty “of the model” as we usually say. Additionally, it may be effective to learn uncertainty estimates based only on g(x), so that the predictions are not a measure of uncertainty of the prediction, but rather it predicts how hard the task is. Do the authors think these properties are actually limitations? Do the authors have any expectation for how problematic these potential limitations may be?

---

### Official Review · Reviewer_m4Xp · 2024-10-31

**Soundness:** 3
**Presentation:** 4
**Contribution:** 3
**Rating:** 6
**Confidence:** 3

**Summary:**

This paper proposes a supervised learning approach for uncertainty estimation in Large Language Models (LLMs), introducing three key variants: white-box supervised (Wb-S), gray-box supervised (Gb-S), and black-box supervised (Bb-S) methods. The authors formulate uncertainty estimation to predict the quality of LLM responses using labeled datasets, leveraging features extracted from hidden layer activations (white-box features) and entropy or probability-related outputs (gray-box features). They demonstrate how hidden layer activations from one LLM can be used to estimate uncertainty in another LLM's outputs, even when the target LLM is a black box. The method is evaluated across multiple tasks, including question answering, multiple-choice questions, and machine translation, showing improved performance over existing unsupervised benchmarks. The paper also provides theoretical insights into why hidden layers contain uncertainty information and distinguishes between uncertainty estimation and calibration.

**Strengths:**

- **Clear formulation**: The paper is very well written and provides a nice formalization of the uncertainty estimation problem for LLMs, distinguishing it from traditional machine learning uncertainty estimation. Additionally, the authors offer some theoretical insights (Proposition 4.1 and Corollary 4.2) explaining why hidden layer activations contain relevant information for uncertainty estimation and why traditional calibration methods may not suffice for LLMs.

- **Comprehensive evaluation**: The experimental evaluation is thorough, covering multiple tasks, models, and settings (white-box, gray-box, black-box). I am familiar with the dataset but not some of the particular baseline models. Nevertheless, the ablation studies and analysis of different layers' contributions provide valuable insights.

- **Adaptability and versatility**: The method is adaptable to different levels of access to the LLM's internals, making it applicable in various deployment scenarios, including when the target LLM is a closed-source model.

- **Utilization of hidden activations**: Demonstrating that one LLM's hidden states can be used to estimate uncertainty in another LLM's outputs is an interesting and impactful finding.

**Weaknesses:**

- **Insufficient uncertainty taxonomy**: The paper does not clearly distinguish between different types of uncertainty (epistemic vs. aleatoric), which is crucial for understanding what kind of uncertainty is being estimated and how it should be interpreted.

- **Limited theoretical analysis of tool LLM reliability**: In the black-box supervised (Bb-S) setting, the method relies on a secondary LLM to estimate uncertainty. The paper lacks a theoretical analysis of when and why this approach works and potential failure modes if the tool LLM itself is uncertain or hallucinating.

- **Scope of evaluation**: The experiments focus on smaller open-source models (7B-13 B parameters). Discussion on the method's scalability to larger models or different architectures (e.g., GPT-3.5, GPT-4 or even larger LLaMA models) is limited, and this has to be clearly highlighted in the limitations and future work.

- **Potential for recursive uncertainty**: The paper does not address the potential issue of recursive uncertainty when the tool LLM used for estimation introduces its own uncertainties or errors, especially in the black-box setting.

- **Limitations discussion**: The limitations section is brief. A more thorough discussion of potential failure modes, edge cases, and the impact of labeled data quality would provide a balanced view of the method's applicability.

**Questions:**

- **Clarification on types of uncertainty**: Can you clarify whether your method primarily addresses epistemic uncertainty (model uncertainty) or aleatoric uncertainty (data uncertainty)? How does this distinction impact the interpretation of the uncertainty scores in your experiments? I suppose the method is more focused on epistemic uncertainty, but it would be helpful to make this explicit.

- **Reliability of the tool LLM in Bb-S setting**: In the black-box supervised (Bb-S) method, what measures are taken to ensure that the tool LLM does not introduce its own uncertainties or hallucinations when estimating the target LLM's uncertainty? Have you considered the impact of the tool LLM's performance on the overall uncertainty estimation?

- **Comparison with alignment-based methods**: How does your approach differ from existing alignment-based methods aiming to improve LLM reliability and uncertainty estimation? Can you elaborate on the unique contributions and advantages of your method? I am quite familiar with alignment in general and uncertainty estimation in ML, so I am trying to better understand the novelty of your approach.

- **Scalability to larger models**: Have you investigated how your method scales to larger LLMs or those with different architectures? Are there any challenges or limitations when applying your approach to models like GPT-3 or GPT-4?

- **Potential failure modes and limitations**: Can you discuss potential scenarios where your method might not perform well, such as adversarial inputs or in low-resource settings? How do the quality and size of the labeled dataset affect the performance of the uncertainty estimation model?

- **Consideration of conformal prediction methods:** You briefly refer to conformal prediction in line 145 and Appendix A. Have you considered using it in your experiment similar to [1]? They can especially help with the calibration of uncertainty estimates (albeit more calibration in the conformal sense than how you describe it in your paper).

**Reference**
- [1] [Large language model validity via enhanced conformal prediction methods](https://arxiv.org/abs/2406.09714)

---

### Official Review · Reviewer_icYD · 2024-11-04

**Soundness:** 2
**Presentation:** 3
**Contribution:** 2
**Rating:** 3
**Confidence:** 4

**Summary:**

This paper discusses a method for predicting the correctness of an LLM output. This is done by first, using a target LLM to generate responses $y$ to prompts $x$, using a scoring function to label the correctness of the output with $s(y, y_{\text{true}})$, generating features $v$ of the prompt-response pair $(x,y)$ with black-box, grey-box or white-box features, then training a model to predict $s(y, y_{\text{true}})$ given $v$.

**Strengths:**

- Uncertainty for language models is still an unsolved problem
- Writing is mostly clear and easy to follow

**Weaknesses:**

- The definition of uncertainty that this work assumes and discusses their method with is debatable. The uncertainty producing module, the $g$ function, is completely modular and external to the model it can be used to evaluate. Therefore, the notion of uncertainty is independent of the model, and only dependent on responses (which can come from any model). However, predictive uncertainty is understood as the "uncertainty in making a prediction". As a thought experiment, suppose a language model is given an input $\textit{What's the capital of France}$ and deterministically outputs $\textit{Berlin}$. Since the model output is deterministic, one would perceive this model as being "incorrect with 0 uncertainty". However, the method in this paper would likely mark this response as "incorrect" and thus label this response as being "highly uncertain". It is also debatable which signal is more useful or actionable - "how uncertain the model is in producing a prediction" vs. "the predicted correctness of a response". I believe these conflicting definitions/notions need to be reconciled.
- While the paper presents the method and motivation in a slightly different manner, at its core, I don't see how the presented method is very different from Azaria & Mitchell [1]. Their method is essentially identical to the "white-box" method this paper presents. This paper extends this core idea to the black and grey box setting, but the idea of training an external model with extracted features to predict some label of correctness is identical. [1] frame their problem in the context of hallucination, whereas this paper's framing is w.r.t. uncertainty, but the idea doesn't differ very much.
- Why wasn't [1] included in the baseline methods for empirical evaluations? Suppose you were to include it - how would it differ from the proposed method in terms of implementation?

[1]: Azaria & Mitchell, 2023: "The Internal State of an LLM Knows When It’s Lying"

**Questions:**

N/A

---

### Official Review · Reviewer_6U6p · 2024-11-04

**Soundness:** 2
**Presentation:** 3
**Contribution:** 2
**Rating:** 3
**Confidence:** 4

**Summary:**

This paper addresses uncertainty estimation and calibration for large language models (LLMs). It introduces a supervised learning method that utilizes labeled datasets to estimate the uncertainty of LLMs’ responses. The authors propose leveraging hidden layer activations of a white-box tool LLM to capture uncertainty information, demonstrating that these activations can improve uncertainty quantification, especially for variable-length responses. The paper presents the method's adaptability across black-box, grey-box, and white-box models and evaluates it on tasks like question-answering and machine translation. The experimental results indicate that this supervised approach outperforms unsupervised methods, showing potential robustness and transferability across in-distribution and out-of-distribution settings.

**Strengths:**

By focusing on uncertainty quantification, the work addresses a crucial aspect for deploying LLMs in real-world applications, especially where reliability and accuracy are paramount.

**Weaknesses:**

1.	The proposed model captures total uncertainty defined in eq (1) but does not distinguish between aleatoric (data-based) and epistemic (model-based) uncertainty. Due to this reason, the proposed method is not sensitive to OOD data as shown in section 5.3, which should be detected using epistemic uncertainty.

2.	From my understanding, the discussion provided in 4.2 cannot explain how uncertainty quantification (UQ) differs between traditional machine learning models and LLMs. Proposition 4.1 only states that the populational minimizers $f^*$ has the conditional independence structure (or sufficient statistics), but all models are learned from empirical, potentially violating this independence in practice. Actually, there are some existing work  using the hidden activations to improve uncertainty estimation, see

Shen, Maohao, Yuheng Bu, Prasanna Sattigeri, Soumya Ghosh, Subhro Das, and Gregory Wornell. "Post-hoc uncertainty learning using a dirichlet meta-model." In Proceedings of the AAAI Conference on Artificial Intelligence, vol. 37, no. 8, pp. 9772-9781. 2023.

Therefore, the difference in the loss function or the presence of conditional independence cannot be the fundamental distinction between uncertainty estimation in traditional ML models and LLMs.

3.	The baselines chosen for comparison may be too simplistic or lack relevance for the current problem scope. Please consider comparing with the following two papers in the rebuttal.

Hou, Bairu, Yujian Liu, Kaizhi Qian, Jacob Andreas, Shiyu Chang, and Yang Zhang. "Decomposing uncertainty for large language models through input clarification ensembling." ICML 2024.

Shen, Maohao, Subhro Das, Kristjan Greenewald, Prasanna Sattigeri, Gregory Wornell, and Soumya Ghosh. "Thermometer: Towards Universal Calibration for Large Language Models." ICML 2024.

**Questions:**

How does using different tool LLMs affect performance? Is a model trained with a more complex tool LLM transferable to a simpler tool LLM?

---

### Note · Authors · 2024-11-28

**Comment:**

We appreciate the reviewers' time and detailed feedback, but we have decided to withdraw this paper from consideration at the conference. Thank you again for your thoughtful comments.

**Withdrawal Confirmation:**

I have read and agree with the venue's withdrawal policy on behalf of myself and my co-authors.